# A spatio-temporally constrained gene regulatory network directed by PBX1/2 acquires limb patterning specificity via HAND2

Marta Losa[1,13], Iros Barozzi[2,13], Marco Osterwalder[3,4,5], Viviana Hermosilla-Aguayo[1], Angela Morabito[6], Brandon H. Chacón[1], Peyman Zarrineh[7], Ausra Girdziusaite[6], Jean Denis Benazet[1], Jianjian Zhu[8], Susan Mackem[8], Terence D. Capellini[9,10], Diane Dickel[3], Nicoletta Bobola[7], Aimée Zuniga[6], Axel Visel[3,11,12], Rolf Zeller[6,14] & Licia Selleri[1,14] ✉

A lingering question in developmental biology has centered on how transcription factors with widespread distribution in vertebrate embryos can perform tissue-specific functions. Here, using the murine hindlimb as a model, we investigate the elusive mechanisms whereby PBX TALE homeoproteins, viewed primarily as HOX cofactors, attain context-specific developmental roles despite ubiquitous presence in the embryo. We first demonstrate that mesenchymal-specific loss of PBX1/2 or the transcriptional regulator HAND2 generates similar limb phenotypes. By combining tissue-specific and temporally controlled mutagenesis with multi-omics approaches, we reconstruct a gene regulatory network (GRN) at organismal-level resolution that is collaboratively directed by PBX1/2 and HAND2 interactions in subsets of posterior hindlimb mesenchymal cells. Genome-wide profiling of PBX1 binding across multiple embryonic tissues further reveals that HAND2 interacts with subsets of PBX-bound regions to regulate limb-specific GRNs. Our research elucidates fundamental principles by which promiscuous transcription factors cooperate with cofactors that display domain-restricted localization to instruct tissue-specific developmental programs.

Genetic studies in the mouse have led to significant insights into the genetic pathways that direct limb bud patterning and morphogenesis in development, evolution, and congenital disease[1,2]. We previously reported that the gene family encoding PBX1/2/3 homeodomain transcription factors of the TALE superclass are essential limb regulators[3,4] that execute hierarchical, overlapping, and iterative functions during limb development[4–7]. PBX1/2/3 are required in limb bud positioning and formation, limb axes establishment, as well as patterning and morphogenesis of limb and girdle skeletal elements, through the control of effector genes such as those encoding SHH in the limb bud posterior mesenchyme[6] and ALX1 in pre-scapular domains[8].

PBX TALE transcription factors have long been regarded as cofactors that increase the low DNA-binding specificity of HOX proteins[3]. However, it remains challenging to envision: (1) how PBX homeoproteins with widespread distribution in the vertebrate embryo

confer functional specificity to HOX proteins that display domain-restricted localization, and (2) how they themselves attain tissue-specific developmental functions. We previously demonstrated that PBX homeoproteins do not act solely as HOX cofactors in limb buds[3], but control 5′ *HoxA/D* gene expression[6]. In particular, homozygous loss of *Pbx1* (*Pbx1−/−*) causes malformations of girdles and stylopod skeletal structures[9], while loss of *Pbx2* or *Pbx3* does not yield limb phenotypes[10,11]. In contrast, compound constitutive loss-of-function of *Pbx1/2* results in multiple developmental abnormalities, including distal limb defects with loss of posterior digits[6], and exacerbates the proximal limb phenotypes reported in *Pbx1−/−* embryos[9]. Since mouse hindlimb buds express very low levels of *Pbx3*[12] and *Pbx4* (the last known *Pbx* family member) is not expressed during limb development[13], compound loss of *Pbx1/2* likely achieves an overall PBX-null state in the developing hindlimb. Accordingly, *Pbx1/2* mutant hindlimbs display more pronounced phenotypes than those observed in forelimbs, which express relatively higher levels of *Pbx3*[12]. However, germline deletion of both *Pbx1* and *Pbx2* results in embryonic lethality by embryonic day (E)10.5[6] due to cardiovascular defects[14], preventing a mechanistic understanding of gene function during limb bud development.

The establishment of *cis*-regulatory landscapes and gene regulatory networks (GRNs) that control limb patterning and outgrowth has provided additional layers to our understanding of this process[15]. Dissection of the chromatin organization and regulation of the murine *Hox* gene clusters[16,17] has contributed mechanistic insight into the roles of *Hox* genes in determining regional identities along the body axis of bilaterians and during limb bud patterning[18]. Specifically, expression of all four *Hox9* paralogs is required for anterior-posterior (AP) polarization of the limb bud and initiation of SHH signaling[19]. We have shown that *Shh* expression in the limb zone of polarizing activity (ZPA) requires PBX1/2[6] and the bHLH transcription factor HAND2[20]. *Hand2* antagonizes *Gli3* expression to establish AP asymmetry and pentadactyly[20,21]. Establishment of the posterior domain of *Hand2* expression is not only dependent on its mutual antagonistic interaction with *Gli3* and early-wave *Hox* genes, but also on HOX-interacting MEIS1/2 transcription factors that bind to a *Hand2* limb enhancer[22,23]. Furthermore, a HAND2-dependent GRN controls compartmentalization of the limb bud mesenchyme, emphasizing the conserved role of HAND2 upstream of SHH[24] during early limb development. Lastly, dissection of the *Gremlin1* (*Grem1*) *cis*-regulatory landscape, which directs *Grem1* expression in limb buds, has provided insights into the robustness and plasticity of distal limb development[25].

While both PBX1/2 and HAND2 independently activate *Shh* expression by interacting with the ZRS distal limb enhancer[6,20,26], it is unknown whether they converge on regulating SHH signaling. Potential genetic interactions between PBX1/2 and HAND2 in GRNs jointly controlled by these factors remain elusive. Here, we combined in vivo tissue-specific and temporally controlled gene inactivation with transgenic and multi-omics approaches using dissected mouse embryonic hindlimb buds to reconstruct a multi-layered GRN of limb regulators collaboratively directed by PBX1/2-HAND2. We established that genetic interaction of *Pbx1* with *Hand2* in limb mesenchyme is required for normal hindlimb development. Further, comparative analyses of PBX genome-wide occupancy in vivo across multiple embryonic tissues uncovered that PBX transcription factors indiscriminately occupy a vast pool of common genomic loci during development. However, PBX-binding gains restrained functionality via cooperative and direct interactions with HAND2, a critical regulator of limb AP asymmetry and pentadactyly. Together, these studies allowed the reconstruction of a spatio-temporally constrained GRN at an organismal- and tissue-level resolution. Broadly, this research shows how promiscuous transcription factors attain context-specific developmental functions via cooperative interactions with select cofactors that enable tissue specificity during vertebrate organogenesis.

## Results

### Early requirement of *Pbx1* and *Pbx2* in hindlimb buds
Given the dominant role of PBX1 among PBX family members during hindlimb development[6], we examined its spatio-temporal distribution from hindlimb bud initiation (E9.0) to digit development (E12.0). Following its expression in the lateral plate mesoderm (E8.0)[6], PBX1 was detected in most limb mesenchymal progenitors and in the apical ectodermal ridge (AER) at the onset of hindlimb bud development (E9.0–10.5, Fig. 1a–c′), while it was restricted to the proximal-most mesenchyme from E11.0 onward (Supplementary Fig. 1a–c). To circumvent early embryonic lethality of *Pbx1/2* compound constitutive null embryos and to decipher tissue-specific *Pbx* functions, we conditionally inactivated *Pbx1*[27] on a *Pbx2*-deficient background[10]. This results in an overall *Pbx*-null state in hindlimb buds, where *Pbx3* expression is extremely weak and restricted to a limited number of cells[12].

AER-specific deletion of *Pbx1* on a *Pbx2*-deficient background using a *Msx2Cre* deleter line[28,29], verified by immunohistochemistry, did not cause limb skeletal defects (Supplementary Fig. 1d–j and Supplementary Table 1), establishing that PBX1/2 are dispensable for AER formation and maintenance. In contrast, mesenchyme-specific deletion of *Pbx1* on a *Pbx2*-deficient background using the *Hoxb6Cre* deleter line[30] (Supplementary Fig. 1j and Supplementary Table 1) resulted in drastic limb abnormalities (Fig. 1d–h′) that phenocopy the limb skeletal defects of constitutive compound *Pbx1/2* mutants[6]. Although the phenotypes of *Pbx1f/f;Pbx2+/−;Hoxb6Cre/+* (hereafter *Pbx1cKOMes;Pbx2+/−*) (Fig. 1e, f) and *Pbx1f/f;Pbx2−/−;Hoxb6Cre/+* (hereafter *Pbx1cKOMes;Pbx2−/−*) (Fig. 1g, h) hindlimbs displayed variable penetrance, the pelvic girdle was dysmorphic and the proximal as well as distal elements were malformed, hypoplastic, or absent in all mutant embryos at E14.5 (Supplementary Table 1). As expected, phenotypes were more severe when both *Pbx1* and *Pbx2* were inactivated, varying from hypoplastic autopodia (Fig. 1g′) to autopod agenesis (Fig. 1h′). Assessment of the temporal requirements of *Pbx1* during hindlimb development using the inducible *Hoxb6Cre* (*Hoxb6CreERT*) deleter line[31], through tamoxifen injections at E8.5, E9.5, E10.0 and E10.5, respectively, showed that *Pbx1/2* are required in the bud mesenchyme for hindlimb patterning prior to E10.5, considering that gene inactivation using this line takes 12–18 h (Fig. 1i, j and Supplementary Fig. 1k). Indeed, inactivation before E10.5 resulted in hindlimb abnormalities that mimicked those observed using the *Hoxb6Cre* allele. In contrast, hindlimbs developed normally when tamoxifen was administered at or after E10.5. Altogether, tissue-specific deletion of *Pbx1* on a *Pbx2*-deficient background using different *Cre* deleter lines (Fig. 1d–j and Supplementary Fig. 1d–k) demonstrates that PBX1/2 are dispensable in the AER, but required in the hindlimb mesenchyme until approximately E10-10.5, the time of hindlimb bud initiation and patterning.

### Inactivation of *Pbx1/2* or *Hand2* causes similar limb defects
The distal hindlimb skeletal phenotypes of E14.5 *Pbx1cKOMes;Pbx2−/−* mouse embryos are similar to those reported in embryos lacking *Hand2* in the limb bud mesenchyme (hereafter *Hand2cKOMes*)[20,24] (Fig. 1h, h′, l) and bearing germline deletion of *Shh* (*Shh−/−*)[32], respectively. Given these similarities, we investigated the underlying cellular and molecular alterations. Mesenchymal apoptosis[33] was not increased in early *Pbx1cKOMes;Pbx2−/−* hindlimb buds compared to controls (34 somites, Fig. 1m). However, later in development, apoptosis was increased in the distal-anterior hindlimb bud mesenchyme of *Pbx1cKOMes;Pbx2−/−* embryos (43 and 49 somites, Fig. 1m), consistent with the increased mesenchymal apoptosis in *Shh*- and *Hand2*-loss-of-function limb buds[20,32,34]. Interestingly, *Pbx1/2*-deficient hindlimb buds expressed negligible levels of *Shh*[6]. In both *Pbx1cKOMes;Pbx2−/−* (Fig. 1n, o) and *Hand2cKOMes*[20,24] hindlimb buds the loss of posterior *Shh* was confirmed by RNA in situ hybridization. These results demonstrate that the striking similarities observed in the limb skeletal phenotypes

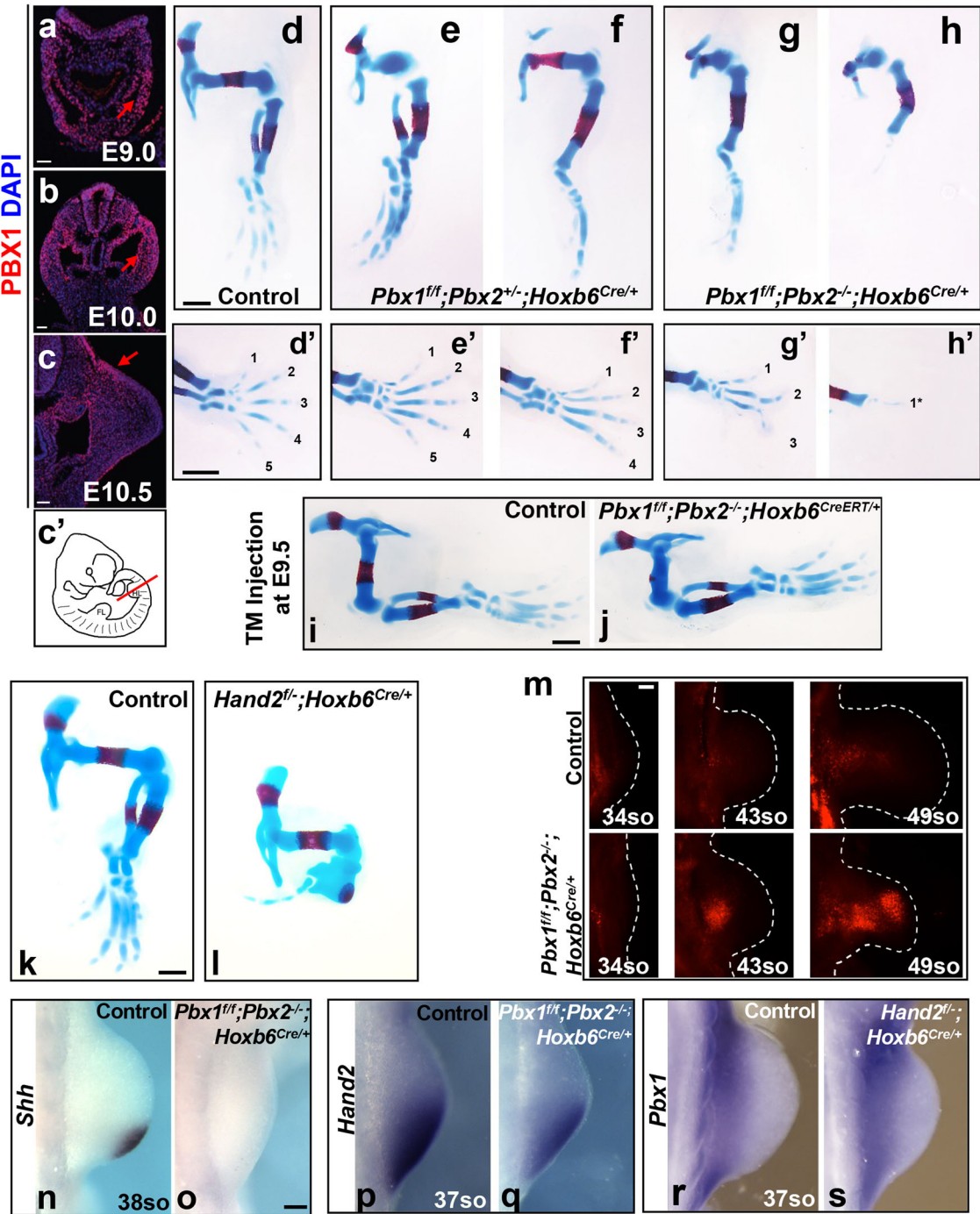

**Fig. 1 | PBX1/2 are required in the hindlimb mesenchyme before E10.5. a–c′** IF showing PBX1 protein (red) distribution in mouse hindlimb buds (HLs) from E9.0 to E10.5. DAPI-labeled nuclei (blue); c′ illustrates plane of section through mouse E10.5 HL. *n* = 3 samples were analyzed per developmental stage. Scale bars: 50 μm. **d−h** Skeletal preparations of E14.5 HLs with limb mesenchyme-specific deletion of *Pbx1* on a *Pbx2*-deficient background using the *Hoxb6Cre* transgene deleter (named *Pbx1cKO^Mes;Pbx2^+/−* and *Pbx1cKO^Mes;Pbx2^−/−*). Cartilage and bone visualized by Alcian Blue and Alizarin Red staining, respectively. *n* = at least 8 samples were analyzed per genotype. Scale bar: 500 μm. **d′−h′** Digits shown from anterior (1) to posterior (5) Scale bar: 500 μm. **i, j** Representative images of *Pbx1^f/f;Pbx2^−/−;Hoxb6^CreERT/+* HL skeletal phenotype compared to *Pbx1^f/f;Pbx2^−/−* control. *n* = at least 8 samples were analyzed per genotype and per developmental stage. Scale bar: 500 μm. Ratios of tibia/femur length in mutant and controls embryos shown in Supplementary Fig. 1k. **k, l** Skeletal preparations of E14.5 HLs lacking *Hand2* in the limb mesenchyme (*Hand2^f/−;Hoxb6^Cre/+*; *n* = 4). Scale bar: 500 μm. **m** Detection of apoptotic cells by LysoTracker Red. *Pbx1^f/f;Pbx2^−/−;HoxB6^Cre/+* and control HLs at E10.25 (34 somites), E11.0 (43 somites), and E11.5 (48 somites). *n* = 3 samples were analyzed per genotype at E10.25 and E11.5; *n* = 2 samples were analyzed per genotype at E11.0. Scale bar: 100 μm. Whole-mount in situ hybridization (WISH) of *Shh* (**n, o**) and *Hand2* (**p, q**) in E10.75 *Pbx1cKO^Mes;Pbx2^−/−* and control HLs. **r, s** WISH of *Pbx1* in E10.75 *Hand2cKO^Mes* and control HLs. In all panels, anterior to the top and posterior to the bottom. *n* = at least 3 samples were analyzed per genotype. Scale bar for (**n−s**): 100 μm; represented by a black bar shown in panel o.

of *Pbx1cKO^Mes;Pbx2^-/-* and *Hand2cKO^Mes* embryos are underpinned by similar cellular and molecular alterations.

Given these results, we asked whether *Pbx1/2* and *Hand2* converge in orchestrating a GRN essential for limb bud development beyond their independent functions in *Shh* regulation. To assess a potential epistatic hierarchy, we examined the spatial expression of *Hand2* in *Pbx1cKO^Mes;Pbx2^-/-* hindlimb buds. Compared to controls, *Hand2* transcript levels were reduced in the proximal-most mesenchyme of *Pbx1cKO^Mes;Pbx2^-/-* developing hindlimbs (Fig. 1p, q), as reported for *Pbx1/2* constitutive mutants[6]. In contrast, *Pbx1* spatial expression was not grossly altered in *Hand2cKO^Mes* hindlimb buds (Fig. 1r, s). Altogether, these findings indicate that *Pbx1* contributes to the regulation of *Hand2* expression, likely as part of a GRN that orchestrates the onset of hindlimb bud development.

## Posterior limb mesenchymal cells co-express *Pbx1/2* and *Hand2*

To investigate the genetic hierarchy and potential convergence of PBX and HAND2 in regulating genes crucial for hindlimb bud patterning, we combined mouse genetics with genome-wide transcriptome and epigenome profiling of dissected hindlimb buds from wild-type mouse embryos at E10.5 (36–38 somites) (Fig. 2a). Unsupervised clustering and marker gene identification from single-cell (sc) RNAseq analysis[35] (Fig. 2b–f and Supplementary Figs. 2, 3) allowed annotation of seven main cell populations within E10.5 hindlimb buds, including mesenchymal cells (*Prrx1*+ and *Meis2*+); epithelial and AER cells (*Epcam*+ and *Wnt6*+); erythrocytes (*Hba-a2*+ and *Klf1*+); endothelial cells (*Pecam1*+ and *Emcn*+); phagocytic cells (*Fcer1g*+ and *Spi1*+); cells characterized by neural lineage markers (*Plp1*+, *Phactr1*+ and *Ascl1*+); and progenitor cells (*Sox2*+ and *Pou5f1*+) (Fig. 2b and Supplementary Fig. 2a–d)[36,37]. The number of clusters identified for each of these populations was stable to parameters' choice, except for mesenchymal cells (Supplementary Fig. 2a). Accordingly, mesenchymal cells were re-clustered across a wide range of resolutions. The resolution value maximizing cluster separation resulted in eight distinct mesenchymal clusters (Fig. 2c, Supplementary Fig. 2b, c and Supplementary Fig. 3a). *Pbx1* and *Pbx2* were detected in both epithelial and mesenchymal cell clusters of the hindlimb bud, although expression was predominant in mesenchymal cells, with *Pbx1* being overall more abundant than *Pbx2* (Fig. 2e, f and Supplementary Fig. 2e). *Pbx3* was either not detected, or detected at markedly lower levels than *Pbx2* in most mesenchymal clusters (Supplementary Fig. 3b, c). Notably, *Pbx3* expression levels were substantially lower in hindlimb than forelimb buds (Supplementary Fig. 3b). In contrast, in the hindlimb bud *Hand2* expression was restricted to a subset of posterior mesenchymal cells (Fig. 2e, f)[20,24]. Additional analyses confirmed that *Hand2* is co-expressed with *Pbx1* and *Pbx2* in a fraction of hindlimb bud mesenchymal cells (orange and red; Fig. 2d), which correspond to mesenchymal clusters 2 and 3 (Fig. 2c, f and Supplementary Fig. 3c). The cells in mesenchymal cluster 3 are characterized by high levels of *Tbx3* and *Isl1* and low levels of *Lhx9* and *Lhx2* transcripts, which identifies them as posterior hindlimb bud mesenchyme (Supplementary Fig. 3d). *HoxA* and *HoxD* gene expression confirmed the spatial assignment of cluster 3 as posterior mesenchymal cells (Supplementary Fig. 3d). In addition, immunofluorescence (IF) on hindlimb bud sections detected PBX1 in most mesenchymal cell nuclei, although levels were lower in the distal domains, while HAND2-positive cells were restricted posteriorly (Fig. 2g–j''). These results demonstrate that nuclear PBX1 and HAND2 transcription factors co-localize in the posterior hindlimb bud mesenchyme.

## A shared PBX-HAND2 GRN in early hindlimb buds

To identify unique and shared transcriptional targets of PBX and HAND2 and their epigenetic landscapes in mouse hindlimb buds, we used chromatin immunoprecipitation followed by sequencing (ChIPseq)[38,39] together with ATACseq (genome-wide assay for

transposase-accessible chromatin using sequencing)[40] in E10.5 hindlimb buds (middle panel, Fig. 2a). From all datasets obtained, only the peaks detected reproducibly across two replicates were analyzed (Supplementary Data 1; see "Methods"). Replicated peaks for PBX1 and HAND2 were merged into one set of 32,691 putative regulatory elements: 6157 bound by both transcription factors; 4536 bound only by HAND2; and 21,998 bound only by PBX1 (Fig. 3a, b and Supplementary Fig. 4a–c). The majority of HAND2 peaks were located distal to the transcription start site (TSS)[24] of annotated genes, while PBX1 peaks were distributed between promoters and TSS-distal regions (Fig. 3a, b). Regions co-bound by both PBX1 and HAND2 predominantly associated with distal elements, including both intergenic and intragenic regions (Fig. 3a). Remarkably, sites co-bound by PBX1 and HAND2 were located in regions of higher chromatin accessibility (Fig. 3c; *p*-value < 2.2e−16, Kruskal−Wallis test) and had greater enrichment values for transcription factor-binding (Fig. 3d, e; *p*-value < 2.2e−16, Mann−Whitney two-sided tests) than sites bound by only one of the two factors. Genomic Regions Enrichment Annotation (GREAT)[41] analysis revealed that the peaks bound by both PBX1 and HAND2 were mostly associated with genes known to regulate limb bud and/or skeletal development (asterisks, Fig. 3f, g), while this association was less apparent or absent for regions bound only by HAND2 or only by PBX1, respectively (Supplementary Fig. 4d–g).

De novo motif discovery (using HOMER)[42] determined whether both PBX1 and HAND2, or only one of them, interacted directly with DNA or as part of transcriptional complexes (Fig. 3h). The HAND2 (annotated by similarity to HAND1::TCF3) and HOX-PBX (TGATTNTT; annotated to HOXC9) motifs were identified as the top two most-enriched motifs in shared peaks, whether they were centered on the PBX1 peak or the HAND2 peak (Fig. 3h; *p*-value < 0.05). Interestingly, the HOX-PBX motif was still the second most enriched sequence motif in peaks bound by HAND2 even in the absence of PBX1 binding. Our de novo motif analysis indicates that PBX1 can bind DNA without HAND2, together with other TALE factors (MEIS1/PREP1), both at promoter and intergenic/intragenic regions (Fig. 3h; *p*-value < 0.05). Notably, the majority of genes associated with PBX1-specific peaks are annotated to developmental processes other than limb morphogenesis (Supplementary Fig. 4f). Moreover, motif analysis using a large set of published binding preferences[42] revealed significant associations of the PBX1-only peaks with binding of other homeobox transcription factors (HOX, MEIS, PDX2, LHX2) at TSS-distal sites (Supplementary Fig. 4h; *p*-value < 1e-5; HOMER). Lastly, co-immunoprecipitation using E10.5 hindlimb buds expressing an endogenous 3xFLAG epitope tagged HAND2 protein[24] (Fig. 3i) and E10.5 wild-type hind- and forelimb buds (Fig. 3j) revealed the presence of PBX1-HAND2 protein complexes in limb buds. Together, these results point to the convergence of PBX1 and HAND2 on *cis*-regulatory modules acting within a GRN that orchestrates hindlimb and digit patterning in the posterior limb bud mesenchyme.

Next, we integrated our sets of PBX1 and HAND2 replicated peaks with H3K27ac (associated with active enhancers) and H3K27me3 (associated with repressed promoters and poised, bivalent enhancers)[43,44] ChIPseq profiles and ATACseq profiles (denoting open chromatin) that we generated from E10.5 hindlimb buds. We also intersected these datasets with published CTCF binding profiles (Supplementary Fig. 5a)[45]. Our analyses revealed that: (1) Intergenic, intragenic and promoter regions co-bound by PBX1 and HAND2 associate with accessible chromatin and H3K27ac enrichment (Supplementary Fig. 5a–c); (2) regions bound only by PBX1 are significantly more accessible and more enriched with H3K27ac than regions bound only by HAND2 (Supplementary Fig. 5a–g); (3) no significant associations are present with genomic regions marked by H3K27me3, with the exception of a fraction of promoter regions bound by PBX1 (Supplementary Fig. 5g); and (4) CTCF binding is not associated with any of the groups analyzed. Taken together, these results indicate that PBX1 and

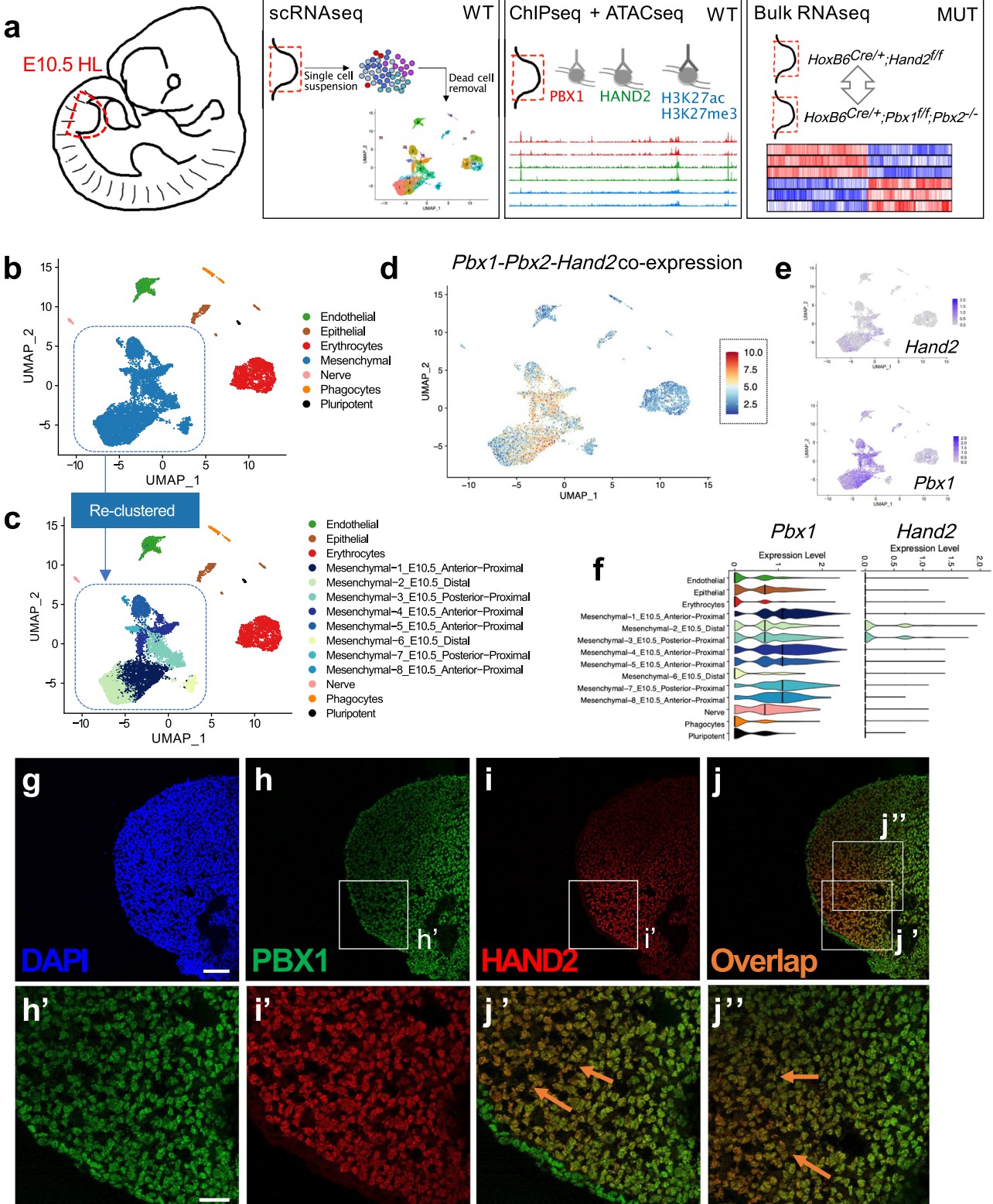

**Fig. 2 | Identification of hindlimb bud mesenchymal subpopulations co-expressing *Pbx1/2* and *Hand2*. a** Workflow of the genome-wide approaches used in parallel. **b**, **c** Uniform manifold approximation and projection (UMAP) representation of 9859 high-quality cells from E10.5–10.75 hindlimb buds (HL), assayed by scRNAseq. **c** Re-clustering of the original mesenchymal subclusters illustrated in (**b**). **d** UMAP highlighting mesenchymal cell populations co-expressing *Pbx1-Pbx2* and *Hand2*. **e**, **f** UMAPs and violin plots displaying normalized expression patterns for *Pbx1* and *Hand2* across single cells and different clusters. Y axes in violin plots are at saturation and differ in scale (panel **f**). **g**–**j** IF of PBX1 (green) and HAND2 (red) protein distribution in E10.5 HL mesenchyme (*n* = 2). Scale bar: 100 μm. Higher magnifications (**h′**–**j″**) show co-localization (orange) in the posterior mesenchyme (orange arrows in **j′** and **j″** point to double positive cells). DAPI-labeled (blue) nuclei. Scale bar: 20 μm.

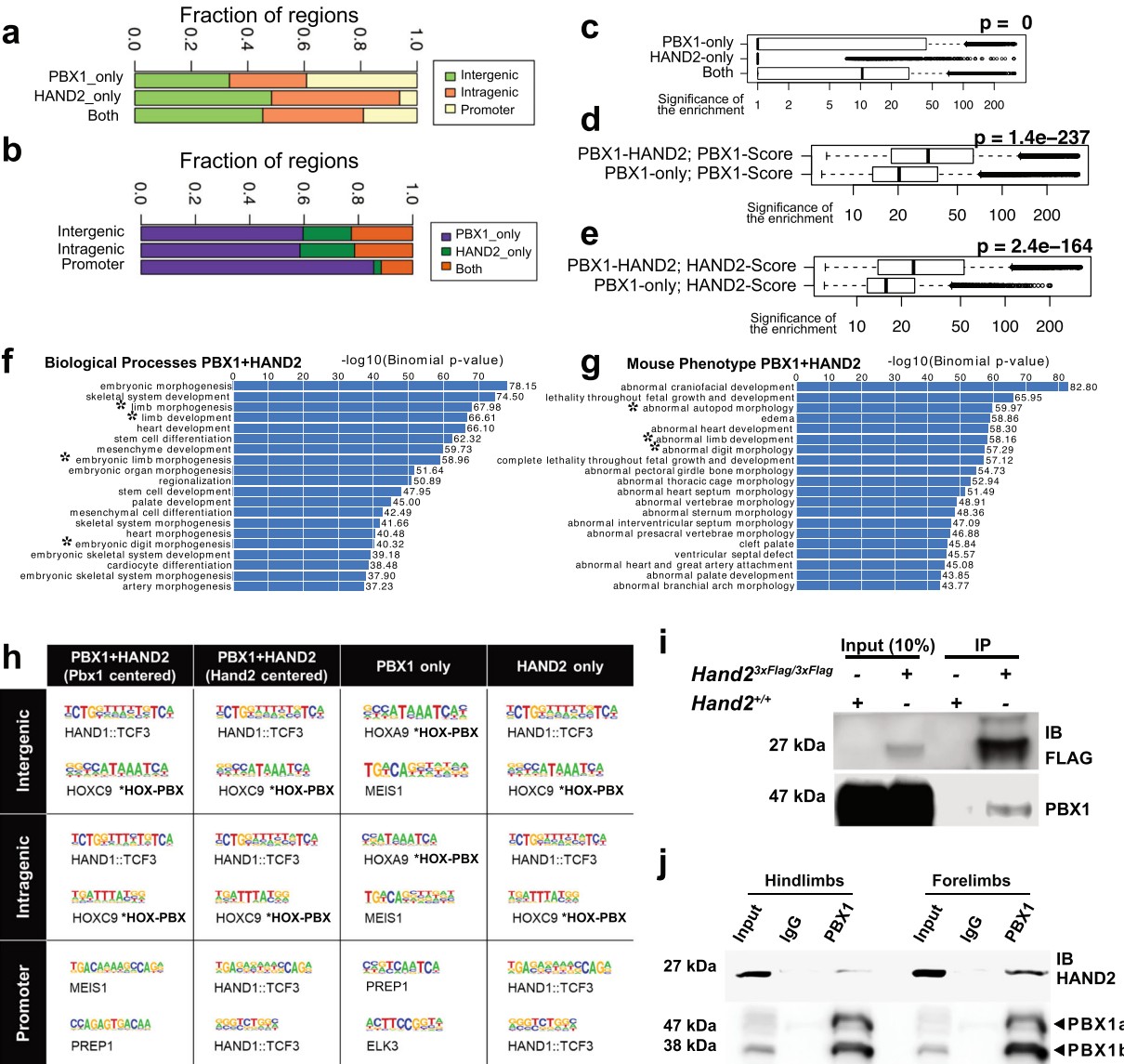

**Fig. 3 | PBX1 and HAND2 control a shared GRN in early hindlimb buds.**
**a** Fractions of PBX1-only, HAND2-only, and PBX1-HAND2 co-bound peaks (both) relative to intergenic, gene promoter, or intragenic regions. **b** Similar to (**a**), fractions of intergenic, promoter, and intragenic regions relative to PBX1-only, HAND2-only, or PBX1-HAND2 co-bound peaks. **c** Chromatin accessibility for PBX1-only and HAND2-only bound peaks and PBX1-HAND2 co-bound peaks. The X axis shows −log10(p-value) of ATACseq enrichment on logarithmic scale. Box plots indicate median, interquartile values, range, and outliers. *P*-value from Kruskal–Wallis test. **d**, **e** Enrichment values for PBX1-only and HAND2-only bound peaks and PBX1-HAND2 co-bound peaks. The X axis shows the −log10(p-value) of ChIPseq enrichment on logarithmic scale. Box plots indicate median, interquartile values, range, and outliers. *P*-values from Mann–Whitney Two-sided tests. Top enriched biological processes (**f**) and mouse phenotypes (**g**) associated with the PBX1-HAND2 shared peaks. *X* axes show the −log10 of uncorrected *p* values. Asterisks* highlight

categories linked to limb development. *P*-values from Binomial Tests. **h** Top two most significantly enriched motifs identified by de novo motif enrichment analysis of genomic regions co-bound by PBX1-HAND2, PBX1-only, or HAND2-only (*q*-value ≤ 0.05). Identity of the most similar annotated motif indicated below each consensus sequence. **i** Co-immunoprecipitation of PBX1 and HAND2-3xFlag in E10.5 hindlimb buds (HL) from endogenously tagged *Hand2^{3xFlag/3xFlag}* mouse embryos. Wild-type (*Hand2^{+/+}*) HLs used as negative controls. Co-immunoprecipitation reveals direct interaction of the endogenous HAND2^{3xFLAG} protein with PBX1 in E10.5 HLs isolated from *Hand2^{3xFlag/3xFlag}* mouse embryos (*n* = 3 biological replicates). Wild-type (*Hand2^{+/+}*) HLs used as negative control. **j** Immunoprecipitation of PBX1-HAND2 protein complex in E10.5 wild-type HL and forelimb buds using PBX1 Ab (*n* = 2 biological replicates). Immunoprecipitation using IgGs served as negative control. kDA kilodalton, IB immunoblot; PBX1a and PBX1b are two known protein isoforms[3].

HAND2 interact with a common set of candidate *cis*-regulatory modules (CRMs) in accessible and active chromatin within the genomic landscapes of limb regulator genes (see below).

**PBX1 regulates *Hand2* via specific CRMs in hindlimb buds**
As *Pbx1/2* controls *Hand2* expression in the proximal-posterior limb bud mesenchyme, wherein these factors co-localize (Fig. 1p, q and Fig. 2d–j″), we screened the *Hand2* topological associating domain (TAD)[46] for limb enhancers by intersecting the PBX1, HAND2, H3K27ac

ChIPseq and ATACseq datasets (Fig. 4a; see "Methods"). We uncovered 17 PBX1-bound putative CRMs with predicted enhancer activity, including 3 known limb enhancers (mm1687, mm1688, and mm1689; Fig. 4; Supplementary Table 2)[23]. Transgenic *LacZ* reporter assays were conducted in E10.5 and E11.5 mouse embryos for the remaining putative PBX1-bound *Hand2* limb enhancers (*n* = 14). These analyses revealed the presence of 7 additional tissue-specific enhancers in the *Hand2* TAD with activity in limb buds and/or other embryonic tissues (Fig. 4d, f, g and Supplementary Fig. 6; Supplementary Table 2).

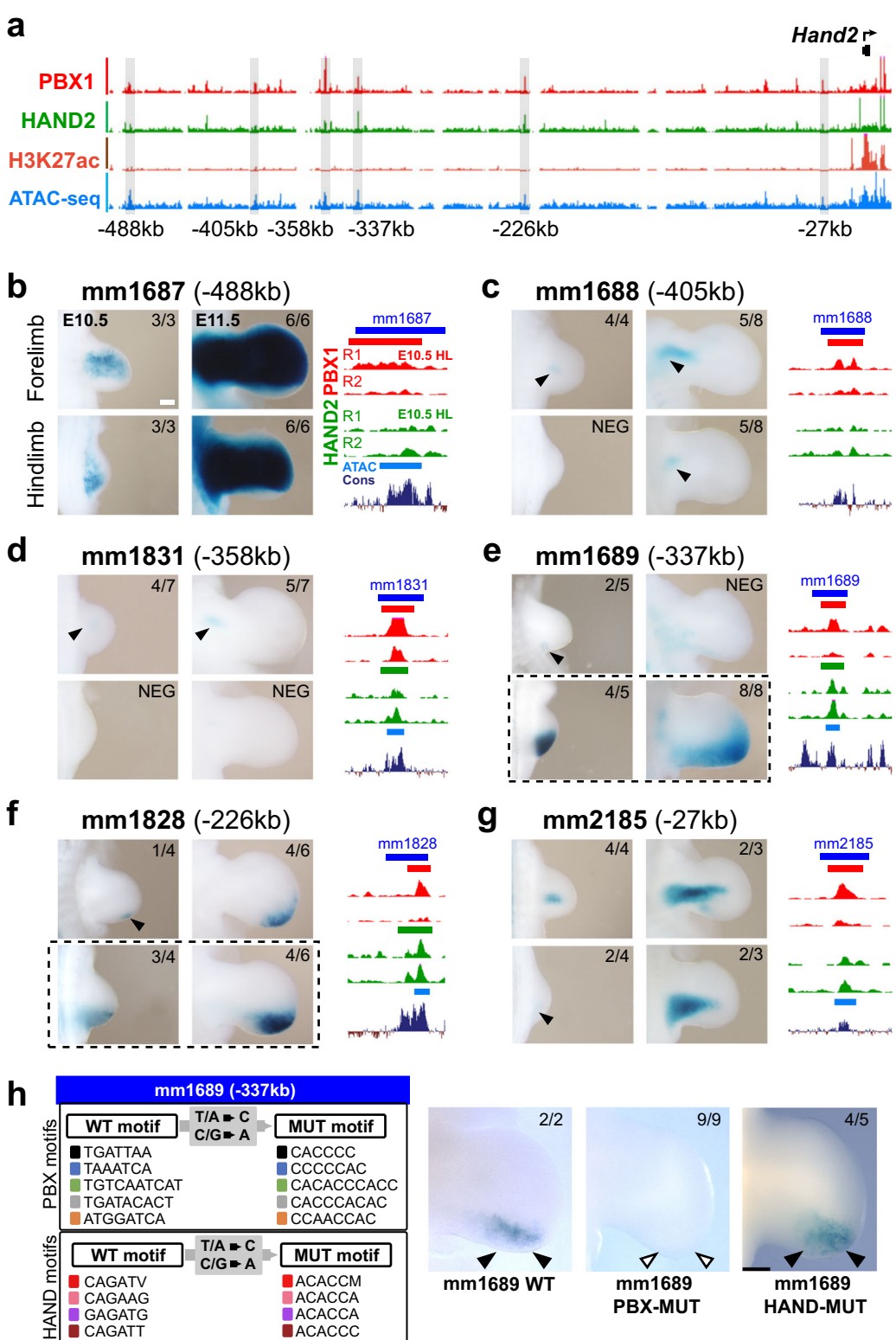

Of these, 3 showed reproducible activities in the *Hand2*-expressing limb bud domain, expanding the number of *Hand2* limb enhancers to 6 in total, all of which were active at E10.5 and E11.5 (Fig. 4b–g and Supplementary Fig. 6). However, only 2 of these enhancers, located 337 kb (mm1689) and 226 kb (mm1828) upstream of the *Hand2* TSS, respectively, displayed activity restricted to the posterior hindlimb bud mesenchyme (Fig. 4e, f; dashed rectangles) overlapping endogenous *Hand2* expression. Interestingly, it was recently reported that

deletion of mm1689 causes loss of *Hand2* expression in mouse limb buds[22]. To assess whether the activity of this CRM is regulated by PBX and/or HAND2, we mutagenized all 'PBX' and 'PBX-HOX' or 'HAND' binding sites within this element (Fig. 4h; see Supplementary Methods) by disrupting the conserved core of the respective binding motifs[3,47]. We then assayed mutagenized (PBX-MUT or HAND-MUT) and wild-type (WT) versions of this enhancer using enSERT[48], a CRISPR/Cas9-mediated site-specific transgenic mouse assay. The activity of

**Fig. 4 | PBX1 is essential for the activity of specific CRMs in the *Hand2* TAD.**
**a** Genomic signatures of candidate limb enhancer elements (gray bars) within the *Hand2* TAD. *Hand2* transcriptional start site (TSS) indicated by arrow (top-right). Enhancers listed according to their relative distance from the *Hand2* TSS (bottom). **b**–**g** Left panels: mouse *LacZ* transgenic reporter assays used to assess the transcriptional enhancer activity (blue staining) in fore- (top panels) and hindlimb buds (HL) (bottom panels) at E10.5 and E11.5. Weak or restricted activity domains highlighted by black arrowheads. The numbers in the top-right corners of all panels show the reproducibility of *LacZ* reporter activity as the number of embryos with limb bud activity over all transgenic founder embryos (n = x/y). Right panels: UCSC genome browser tracks depicting the called PBX1 (red) and HAND2 (green) replicated ChIPseq peaks in E10.5 HLs for each element tested (blue). Called ATACseq peaks (azure) and placental conservation (Cons, dark purple) are also shown. Six enhancers display bona fide limb activity in one or both developmental stages

analyzed. Corresponding Vista enhancer IDs (mm: mus musculus) indicated. Scale bar: 200 μm; represented by a white bar in panel 4b. **h** Analysis of enhancer activity for wild-type (WT) and mutant (MUT) versions of the mm1689 enhancer in E11.5 HLs using enSERT transgenesis. (Left) Strategy to mutagenize all PBX (PBX-MUT) or HAND (HAND-MUT) binding sites. (Right) Activity of PBX-MUT enhancer is lost, while activity of HAND-MUT enhancer remains grossly unperturbed. The differences in X-gal staining of the WT mm1689 enhancer are a likely consequence of the transgenesis technique used (e, random insertion; h, targeted insertion into the H11 locus by enSERT[48]). The numbers in the top-right corners of all panels show reproducibility of the limb bud *LacZ* reporter activity (black arrowheads) or lack thereof (empty arrowheads) over all transgenic founder embryos (n = x/y). Only transgenic embryos carrying at least two copies of the reporter transgene correctly inserted at the H11 locus[48] were included (see Methods). Scale bar: 200 μm; represented by a black bar in panel mm1689 HAND-MUT.

enhancer mm1689 was lost when the PBX-binding motifs were mutagenized (n = 9/9), while it was maintained when the HAND motifs were mutated (n = 4/5) (Fig. 4h). Together, these results demonstrate that PBX, but not HAND2, binding sites are essential for the activity of this *Hand2* enhancer in mouse hindlimb buds.

## PBX and HAND2 collaboratively control limb regulators

To identify GRNs co-regulated by PBX1/2 and HAND2, we conducted bulk RNAseq on E10.5 *Pbx1cKO^Mes^;Pbx2^−/−^* and *Hand2cKO^Mes^* hindlimb buds compared to littermate controls (Supplementary Fig. 7a, b). Statistical analyses identified 1489 differentially expressed genes (DEGs) in *Pbx1cKO^Mes^;Pbx2^−/−^* and 375 DEGs in *Hand2cKO^Mes^* hindlimb buds. Intersection of both transcriptomic datasets, considering only genes that were detected in all replicates for both transcription factors, identified 46 DEGs significantly upregulated in both types of mutant hindlimb buds, which defines them as genes repressed by both PBX1/2 and HAND2 (Fig. 5a, top-left panel; Supplementary Fig. 7b; Supplementary Data 2). GO analyses of the PBX1-HAND2 repressed transcriptional targets revealed significant associations with 'developmental processes' and 'transcription factors' categories (Supplementary Fig. 7c). In addition, 37 genes were significantly downregulated in both types of mutant hindlimb buds compared to controls (Fig. 5a, top-left panel; Supplementary Fig. 7b, c; Supplementary Data 2), which identifies them as target genes positively regulated by both PBX1/2 and HAND2. Lastly, 31 DEGs were altered in a discordant manner in the two mutant mouse lines, *i.e.* transcript levels were either upregulated in *Pbx1cKO^Mes^;Pbx2^−/−^* and downregulated in *Hand2cKO^Mes^* mutant hindlimb buds, or vice versa (Fig. 5a, top-left panel; Supplementary Fig. 7b, c). The GO term 'transcription factors' was significantly enriched in all DEG groups (Supplementary Fig. 7c; TFs), suggesting that PBX1/2 and HAND2 act as key regulators of a downstream GRN comprising limb transcriptional regulators.

Given that, upon signal transduction, many transcription factors translocate from the cytoplasm to the nucleus where they interact with other transcription factors and/or bind directly to DNA[43], we investigated whether the PBX1/2-HAND2 target genes are co-expressed with *Pbx1/2* and *Hand2* in the same hindlimb bud cells. To this end, we analyzed the mesenchymal clusters in our scRNAseq datasets, based on the statistical threshold 'area under the curve' >=0.55 (see Methods). Remarkably, seven transcription factors with essential functions during limb and/or skeletal development, namely *Msx1*[49], *Alx3*[50], *Lhx9*[51], *Prrx1*[52], *Zfhx4*[53], *Ets2*[54] and *Snai1*[55,56], were identified above this statistical threshold (Fig. 5a, top-right panel, red asterisks). In addition, we evaluated whether the expression of the identified PBX1/2-HAND2 target genes might be skewed to specific mesenchymal cell clusters. This analysis detected highest expression of PBX1/2 and HAND2 target genes in mesenchymal clusters 1 to 4 (Fig. 5a, bottom panels; Fig. 5b, c). An orthogonal analysis using an available computational compendium of *Pbx1* and *Hand2* target genes (Dorothea[57]), corroborated these

results (Supplementary Fig. 7d). Of the clusters with consistently highest expression of PBX1/2 and HAND2 target genes, mesenchymal cluster 3 exhibited highest co-expression levels of *Pbx1/Pbx2* and *Hand2* (Fig. 2d–f). Our analysis also showed that PBX1 and HAND2 protein complexes interact with promoters and intergenic regions located in accessible and active chromatin at loci of shared target genes, including *Prrx1* and *Msx1* (Fig. 5d), which are co-expressed with *Pbx1/2* and *Hand2* in mesenchymal cluster 3. Furthermore, validation by RNAscope[58] of select shared target genes -representative of upregulated and downregulated targets- that had been identified through bioinformatic analysis revealed that spatial expression of *Prrx1* and *Snai1* overlaps *Pbx1*, which is broadly expressed in the hindlimb bud (Fig. 2h), and also *Hand2*, specifically in posterior mesenchymal cells (Fig. 6a). Consistent with the bioinformatic predictions, *Prrx1* expression was upregulated while *Snai1* expression was downregulated or lost in both *Pbx1cKO^Mes^;Pbx2^−/−^* and *Hand2cKO^Mes^* hindlimb buds (Fig. 6b, c). Functional annotation of the identified PBX-HAND2 target genes according to their spatial expression and associated limb and/or skeletal phenotypes (Supplementary Data 3) allowed the inference of a GRN revealing both concordant and discordant regulation of downstream genes with key functions during limb bud development (Supplementary Fig. 8). Notably, this analysis suggests that PBX and HAND2 co-repress genes with roles in chondro-osteogenic differentiation, while in parallel positively regulate genes that inhibit this differentiation process (Supplementary Fig. 8). Together, these results establish that the co-regulated PBX1/2-HAND2 target genes: (1) are significantly perturbed in hindlimb buds with mesenchymal-specific loss of *Pbx1/2* and *Hand2;* (2) are bound by both PBX1 and HAND2; (3) are associated with open and active chromatin; (4) are co-expressed in posterior hindlimb bud mesenchyme at early stages; and (5) inhibit regulators of chondro-osteogenic differentiation, thus preventing the precocious onset of this process.

We then examined whether previously defined TADs[46,59] comprising the identified DEGs exhibited a higher number of putative regulatory elements bound by PBX1 and/or HAND2 than TADs encompassing genes whose expression was not altered in *Pbx1cKO^Mes^;Pbx2^−/−^* or *Hand2cKO^Mes^* hindlimbs (Supplementary Fig. 9). This analysis indicated that distal elements within TADs spanning DEGs identified in either of the two mutants showed a trend toward higher regulatory scores than genes whose expression was not altered (based on both the total number and individual strength of the sites contacted by PBX1 and/or HAND2 within the TAD; see Methods). Moreover, the genes within TADs with the highest number and strength of distal regions bound by PBX1 and/or HAND2 were generally upregulated or regulated in a discordant manner in both mutant hindlimb buds (Supplementary Fig. 9, red arrows). These results indicate that: (1) TADs with higher numbers of DEGs comprise higher numbers of genomic regions bound by PBX1 and/or HAND2; and (2) genes that are repressed by these two transcriptional regulators are embedded within complex *cis*-regulatory landscapes.

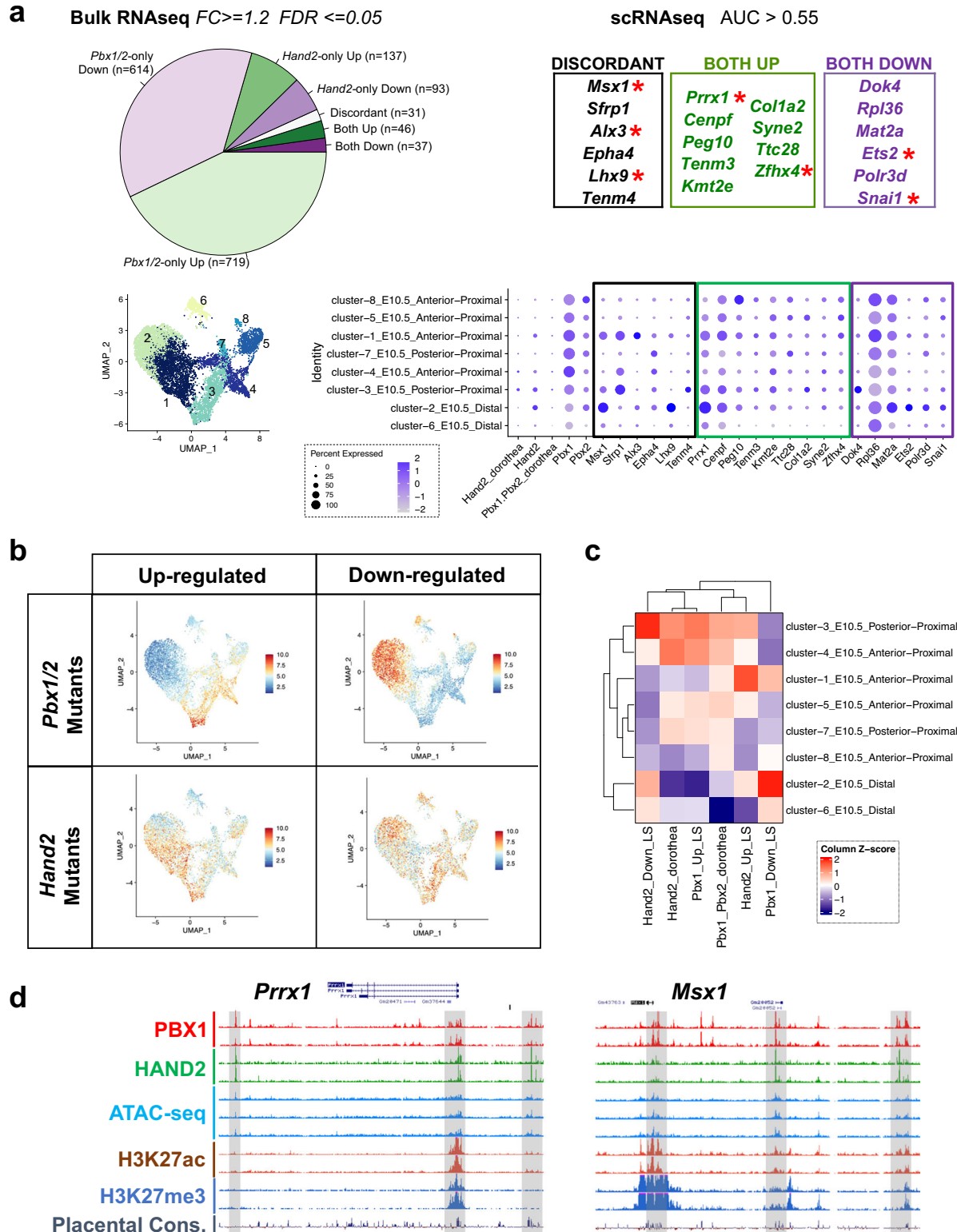

## *Pbx1-Hand2* interaction is essential for hindlimb patterning

We next assessed the developmental impact of the PBX1/2-HAND2-dependent GRN on mouse hindlimb bud patterning and skeletal development by performing a classical genetic interaction experiment. We generated compound mutant embryos lacking *Pbx1* and one allele of *Hand2* in the limb bud mesenchyme using the *Hoxb6Cre* deleter (Fig. 7). The wild-type hindlimb skeletal morphology was maintained in both single heterozygous *Pbx1* and *Hand2* embryos (*Pbx1^f/+^;Hoxb6^Cre/+^* and *Hand2^+/−^;Hoxb6^Cre^*) compared to wild-type controls (Fig. 7a, b). In contrast, pelvic malformations and a shorter femur (*n* = 4/4; Fig. 7g) were detected in *Pbx1^f/f^;Hoxb6^Cre/+^* single homozygous mutant embryos (Fig. 7c). Compound *Pbx1^f/f^;Hand2^+/−^;Hoxb6^Cre/+^* embryos lacking both copies of *Pbx1* and one copy of *Hand2* in the hindlimb bud mesenchyme (*n* = 7; Fig. 7g) revealed a striking genetic

**Fig. 5 | Target genes co-regulated by PBX1/2 and HAND2 are essential for hindlimb bud development. a** DEGs identified by bulk RNAseq in *Pbx1cKO^Mes;Pbx2^-/-* and *Hand2cKO^Mes* hindlimb buds (HLs) *versus* respective controls at E10.5. (Top-left) Pie chart indicating the proportion of DEGs shared between *Pbx1/2* and *Hand2* mutant HLs (Both), or specific to single mutant (*Pbx1/2*-only and *Hand2*-only) HLs. Number of DEGs in each category indicated in brackets. (Top-right) Intersection of bulk RNAseq with scRNAseq. DEGs from both *Pbx1/2*-deficient and *Hand2*-deficient HLs and co-expressed in the same cells as *Pbx1/2* and *Hand2* (AUC > 0.55). Transcription factors with known essential functions in limb bud development highlighted by red asterisk. (Bottom Right) Dot plot showing the expression of the identified target genes in the mesenchymal subclusters established by scRNAseq. Dot sizes represent the proportion of cells within a given population that expresses

the target gene; color intensities indicate average expression levels. (Bottom Left) UMAP of the mesenchymal cells highlighting the clusters. **b** UMAP of the mesenchymal cells in which each cell cluster is color-coded according to the expression of *Pbx1/2 and Hand2* target genes. **c** Heatmap showing the relative (z-score) median expression of different subsets of *Pbx1/2 and Hand2* target genes in the different mesenchymal clusters based on both our datasets and an available computational compendium (Dorothea). **d** UCSC genome browser tracks encompassing the *Prrx1* and *Msx1* loci. Tracks: PBX1 and HAND2 ChIPseq, red and green, respectively; ATACseq, aqua; H3K27ac and H3K27me3, brown and blue, respectively; placental conservation, dark purple. Gray bars highlight predicted CRMs bound by PBX1 and HAND2.

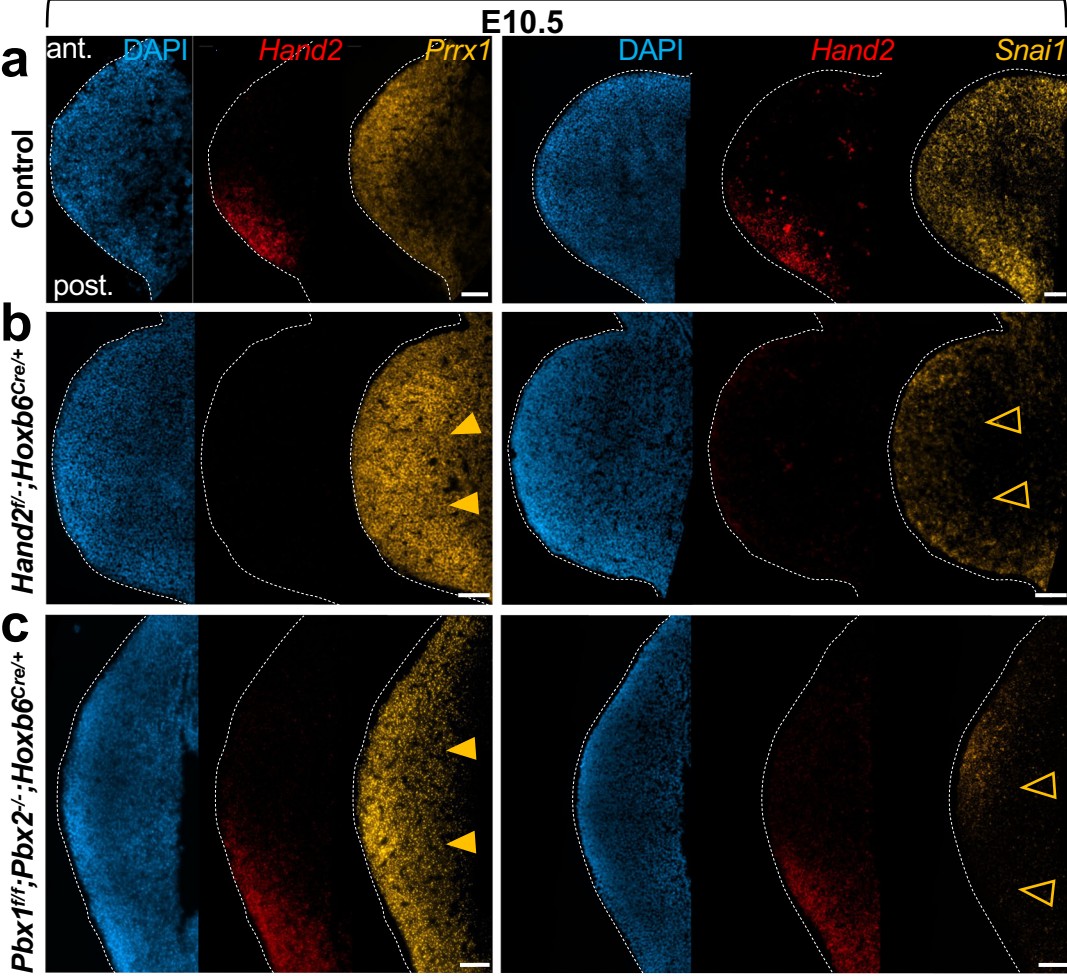

**Fig. 6 | In vivo validation of predicted PBX1/2-HAND2 target genes *Prrx1* and *Snai1* confirms the bioinformatic analysis.** Hindlimb bud (HL) sections from control (**a**); *Hand2^fl/-;Hoxb6^Cre/+* (**b**); and *Pbx1^fl/f;Pbx2^-/-;Hoxb6^CreERT/+* (**c**) embryos at E10.5 were hybridized using probes for *Hand2, Prrx1* and *Snai1* (*n* = 3; see "Methods"). *Hand2* transcripts (red fluorescent dots) are absent from the posterior

mesenchyme in *Hand2^fl/-;Hoxb6^Cre/+* HLs, while *Prrx1* transcripts are upregulated (orange arrowheads) and *Snai1* transcripts downregulated (orange empty arrowheads) in HLs of both mutant genotypes, as predicted by bioinformatic analysis. *Prrx1* and *Snai1* transcripts visualized by yellow fluorescent dots. DAPI-counterstained nuclei (blue). Scale bar: 100 μm.

interaction in hindlimb skeletal and digit patterning. This phenotype comprised a conspicuously dysmorphic pelvis, shorter femur, and severe fibular defects, including agenesis (Fig. 7d–f), in combination with variable reduction or loss of posterior digits (Fig. 7d'–f', g). Despite the variable expressivity of the autopod phenotype in compound *Pbx1^fl/f;Hand2^+/-;Hoxb6^Cre/+* mutants, it is important to note that no autopod defects were observed in either *Pbx1^fl/f;Hoxb6^Cre/+* or *Hand2^+/-;Hoxb6^Cre/+* hindlimbs (Fig. 7g). These results establish that *Pbx1* and *Hand2* genetically interact in the control of hindlimb bud patterning and skeletal development.

### Promiscuous PBX1 acquires limb bud functions via HAND2

Noting that *Pbx* genes are broadly expressed in the embryo and fulfill essential pleiotropic roles during organogenesis[47], we next asked whether context-specific PBX functions are achieved by tissue-specific PBX binding or cooperativity with different transcription factors that confer specific functionality. We therefore compared PBX genome-wide binding profiles across multiple tissues of the developing mouse embryo. Given the essential roles of PBX proteins that we reported in midface morphogenesis[60,61], we generated additional PBX1 ChIPseq datasets from the murine midface (MF) at E10.5 and E11.5. In addition,

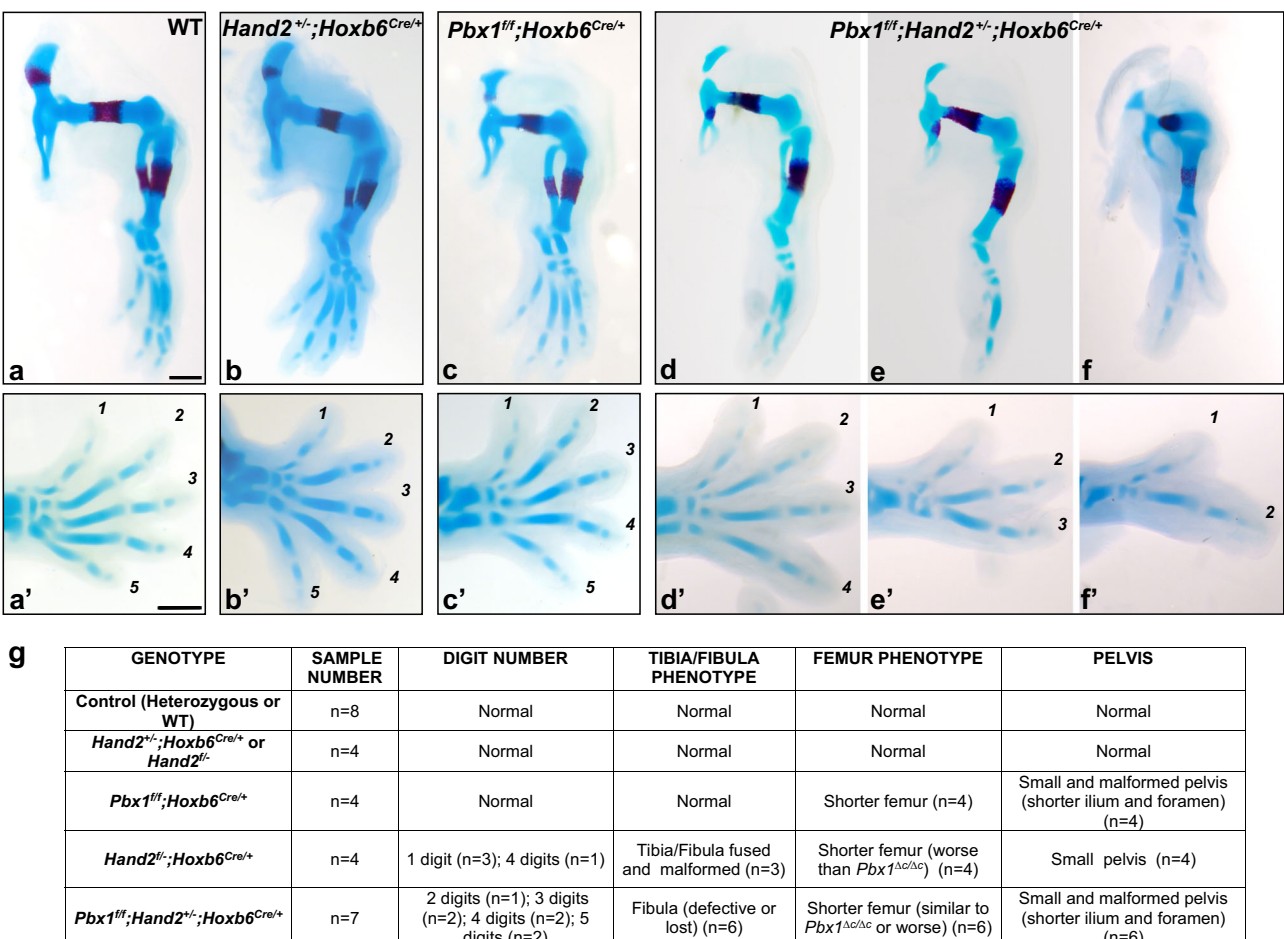

**Fig. 7 | In vivo genetic interaction of *Pbx1* and *Hand2* directs patterning of posterior hindlimb skeletal elements. a–f'** Hindlimb skeletons at E14.5 (blue: cartilage; red: bone). Digits shown from anterior (1) to posterior (5). **a, a'** Representative control embryo (Wildtype, WT) (*n* = 8). (**b, b'**) *Hand2+/-;Hoxb6Cre/+* hindlimb with normal morphology. **c, c'** *Pbx1f/f;Hoxb6Cre/+* hindlimbs displaying pelvic malformations and shorter femur (*n* = 4/4). (**d–f, d'–f'**) *Pbx1f/f;Hand2+/-;Hoxb6Cre/+* hindlimbs exhibiting malformed pelvic bones, shorter or truncated femur, abnormal

or absent fibula, hypoplastic or absent tarsals and metatarsal and variable loss of posterior digits (*n* = 6/7). Distal-most autopod abnormalities (in tarsal, metatarsal, and phalanges) not observed in *Pbx1f/f;Hoxb6Cre/+* or *Hand2+/-;Hoxb6Cre/+* mutants. Scale bar: 500 μm. **g** Table listing the number of embryos with the different genotypes (shown in **a–f**) with hindlimb skeletal defects over the total number of embryos analyzed.

as PBX interacts with MEIS and HOXA2 to drive a transcriptional program in the second branchial arch (BA2)[38], an available PBX1 ChIPseq dataset from E11.5 BA2 was also included in this study. Intersection of the three datasets [PBX1 genome-wide occupancy in the hindlimb bud (HL), MF, and BA2], including only peaks called in both ChIPseq replicates, identified more than 10,000 PBX1-bound genomic regions in each of the tissues analyzed, a large fraction of which was shared by all three embryonic tissues (Fig. 8a; see Supplementary Table 3 for numbers and percentages). The fraction of peaks shared across these datasets is very similar in size to the binding overlap expected across biological replicates of the same genome-wide binding experiment (>50%)[62], pointing to highly promiscuous PBX binding in different embryonic tissues. GREAT analysis of the PBX-binding profiles in HL and BA2 revealed significant enrichment for diverse developmental processes. GO terms associated with limb development (Fig. 8b, c) were included, but they were not among the top enriched processes in the HL -or in the BA2- dataset. Next, we intersected PBX-bound peaks with HAND2 or HOXA2 cistromes[63]. This analysis revealed that HAND2 and HOXA2 each occupied only a small subset of PBX peaks. Focusing on the shared HAND2-PBX1 ChIPseq peaks in HL revealed that "limb development" and "limb morphogenesis" were among the developmental processes with highest enrichment (Fig. 8d, e). Moreover, HAND2 showed preferential interactions with PBX1-bound genomic

regions in HL (48%) compared to MF (18%) tissues (Fig. 8d; Supplementary Table 3). In contrast, HOXA2 exhibited preferential binding to PBX1-bound regions within BA2 (70%) compared to HL (43%) (Fig. 8f; Supplementary Table 3). Comparative analysis of HOXA2 binding between BA2 and MF tissues was not possible, as the MF is a so-called "*Hox*-less" domain[64]. Notably, HAND2- and HOXA2-bound peaks showed only a minimal overlap (12%) and were exclusive (Fig. 8g, Supplementary Table 3). For example, PBX1 binds to similar CRMs in the genomic landscapes of *Prrx1* and *Zfp703* target genes in HL and BA2. However, at the *Prrx1* locus PBX1 binds the majority of shared peaks together with HAND2 in hindlimb buds, while at the *Zfp703* locus PBX1 binds the majority of shared peaks together with HOXA2 in BA2 (Fig. 8h, i). These data suggest that HAND2 and HOXA2 select for different PBX-binding events in distinct domains of expression (HL and BA2, respectively), conferring tissue specificity to PBX function. Lastly, comparing HL to BA2 tissue, we quantified PBX1 binding enrichment of peaks that also overlap with HAND2 or HOXA2 binding. We found that shared PBX1-HAND2 peaks in HL are significantly more enriched relative to PBX1 peaks that do not overlap HAND2 or HOXA2 binding (*p*-value < 2.2e−16) (Fig. 8j). Similarly, co-binding with HOX-A2 significantly increased PBX1 binding in BA2 (*p*-value < 2.2e−16) (Fig. 8j). In summary, these results indicate that promiscuous PBX transcription factors occupy a vast pool of shared genomic loci during

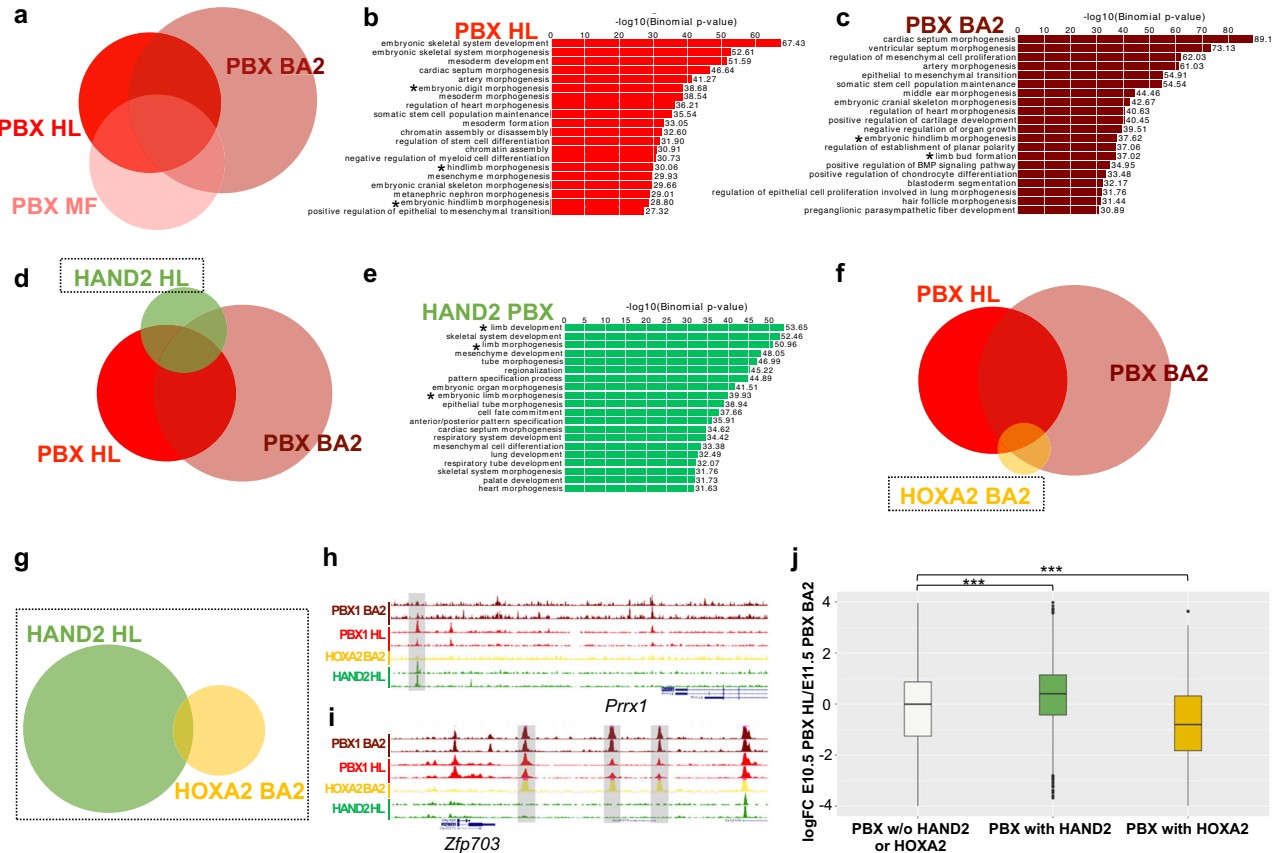

**Fig. 8 | HAND2 selects different subsets of PBX1 peaks to confer early limb patterning functions to PBX1. a–c** Analysis of genome-wide PBX1 (PBX) binding in E10.5 hindlimb bud (HL), E11.5 second branchial arch (BA2) and E11.5 midface (MF). **a** Overlap of PBX-bound peaks in HL, BA2 and MF. Venn diagram highlighting large overlap between PBX-bound loci across different embryonic tissues (for percentages see Supplementary Table 3). Bar plots show the enrichment of GO terms for genes involved in developmental processes (GREAT) that are associated with PBX-bound regions in HL (**b**) and BA2 (**c**). Bar length corresponds to uncorrected *p*-values (X axis values). Asterisks (*) indicate limb development and morphogenesis GO categories. *P*-values from Binomial Tests. **d** Venn diagram of PBX-bound regions in HL and BA2, with HAND2-bound regions in BA2. **e** GREAT analysis of regions co-bound by PBX-HAND2 in HL identifies limb development and limb morphogenesis (*) among the top GO terms. **f** Overlap of PBX-bound regions in HL and BA2, with HOXA2-bound regions in BA2. **g** Venn diagram of regions bound by HOXA2 in BA2 with those bound by HAND2 in HL. **h, i** UCSC genome browser tracks (mm10) showing the ChIPseq peaks from PBX1 BA2 (brown); PBX1 HL (red); HOXA2 BA2 (yellow); and HAND2 HL (green) at the *Prrx1* (**h**) and *Zfp703* (**i**) genomic landscapes. Shared peaks are shaded gray. **j** Box plots of ratio (log2Fold Change; logFC) of normalized PBX1 ChIPseq signals between HL and BA2 at the indicated subsets of PBX-bound regions. Box plots indicate median, interquartile values, range, and outliers. ***P*-value <2.2e−16. *P*-values from Mann–Whitney two-sided tests.

embryonic development, but acquire restrained tissue specificity via cooperation with distinct transcriptional regulators, such as HAND2 in developing limb buds.

## Discussion

The regulation of gene expression, which is critical for tissue patterning and organ morphogenesis, is often mediated by interactions of transcription factors within complexes[65]. Despite decades of research, it remains unclear how spatio-temporal interactions among transcription factors and select cofactors achieve functional specificity and how transcription factor complexes regulate their target genes through CRMs. We previously reported that, despite broad expression of *Pbx* genes during embryonic development, their encoded factors control distinct target and effector genes in different tissues and organs[4,14,47,66–70]. Our research highlighted the essential roles of *Pbx1/2* during limb development[4,47], but did not identify the cell type(s) that require PBX function, nor the time window during which limb development depends on PBX. In addition, our analyses did not reveal how PBX homeoproteins attain functional limb specificity, despite the widespread embryonic expression of their encoding genes. Here, we chose the hindlimb bud as a model, since inactivation of both *Pbx1* and *Pbx2* causes more severe phenotypes in mouse hindlimbs than

forelimbs[5,6]. We established that PBX1/2 are dispensable for AER formation and function, but essential in the mesenchyme for initiation of hindlimb bud development, which points to crucial PBX roles in mesodermal progenitors. Consistent with essential functions in mesoderm patterning, we reported that PBX1/2 control *Polycomb* and *Hox* gene expression in the paraxial mesoderm, which gives rise to the axial skeleton[5]. Notably, it was recently reported that a TALE-HOX code establishes a chromatin landscape permissive for recruitment of the WNT-effector LEF1, which in turn unlocks WNT-mediated transcriptional programs that drive paraxial mesodermal fates[71]. Together, these studies establish that PBX1/2 cooperate with tissue-specific cofactors to execute essential roles in specification and patterning of axial and limb mesodermal lineages.

Mouse embryos lacking either limb bud mesenchymal *Pbx1/2* or *Hand2* phenocopy the morphological and molecular defects of embryonic limbs that lack *Shh*[32] or its limb ZRS enhancer[26], resulting in loss of posterior digits. We and others reported that activation of *Shh* in the posterior limb bud mesoderm is controlled by several transcriptional regulators, including interactions of HOX, PBX, HAND2, and ETS with the ZRS; whereas TWIST1, ETV, and GATA prevent anterior ectopic activation of *Shh* expression[6,20,54,72–75]. Direct interactions of HAND2 with TWIST1 are essential for limb bud development, as their

altered dimerization causes Saethre-Chotzen syndrome with associated distal limb skeletal malformations[76]. In this study, we establish that a PBX1/2-HAND2-controlled GRN regulates the early phase of hindlimb bud patterning and outgrowth. We also show that PBX1 regulates *Hand2* expression by interacting with select CRMs active within the posterior hindlimb mesenchyme. Indeed, mutations of all PBX and PBX-HOX or HAND binding sites in CRM mm1689 revealed that PBX is essential for its activity, while HAND2 is dispensable. A recent study reported that deletion of this enhancer, which also encodes MEIS and HOXD13 binding sites, results in loss of *Hand2* expression in mouse limb buds[22]. These findings suggest that PBX could be part of a large transcriptional complex required for *Hand2* activation. Together, these results provide the first mechanistic evidence for how the loss of *Pbx1/2* or *Hand2* causes similar limb skeletal, cellular, and molecular alterations.

Transcription factors that act within the same GRN must be co-expressed within the same cells[43]. Accordingly, synthetic GRN modeling proposed that co-expression is an indicator of active co-regulation within a given cellular context[77]. Our study establishes that both *Pbx1/2* and *Hand2* are co-expressed within restricted mesenchymal cell populations in the posterior hindlimb bud, where PBX1 and HAND2 proteins show nuclear co-localization. The spatio-temporally constrained PBX1/2-HAND2-dependent GRN consists of essential regulators of limb development that are dysregulated in *Pbx1/2* and *Hand2* mutant hindlimb buds. While the strategy employed in this study to identify a PBX-HAND2 target GRN and relevant regulatory elements points to *cis*-regulation, *trans*-regulation might also play a role within this GRN. Similarly, while shared target genes are regulated in a cell autonomous manner in the mesenchymal cell clusters that co-express *Pbx1/2* and *Hand2*, control of target genes in other clusters through non-cell autonomous mechanisms remains to be explored.

Inactivation of *Meis1/2* genes, which encode TALE proteins that can dimerize with PBX[3,47] and are broadly expressed in the mouse embryo, results in distal limb abnormalities including loss of posterior digits[22,78], similar to those reported in *Pbx1/2*-deficient mice. Furthermore, MEIS and TBX transcription factors control a limb-specific GRN by co-regulating enhancers associated with genes essential for limb bud initiation such as *Fgf10*[22]. While these findings demonstrated that *Hand2* and *Shh* expression is altered in *Meis1/2*-deficient hindlimb buds, it remains to be determined whether MEIS proteins are part of the PBX1/2-HAND2-directed GRN (this study), or of a larger shared GRN. Our de novo motif analysis indicates that in hindlimb buds, where *Hand2* is expressed, PBX1 can bind DNA at promoter and intergenic/intragenic regions without HAND2 and together with other TALE factors such as MEIS1/PREP1, in agreement with ChIPseq studies using whole embryos[79]. Therefore, we can envisage a scenario whereby MEIS proteins, together with PBX1/2 and HAND2, could be part of one large transcriptional complex, possibly with other homeodomain transcription factors that participate in the regulation of shared target genes with essential functions during early limb development. Supporting this view, it was reported that MEIS and PBX largely occupy the same genomic regions in branchial arches[38,63]. Transcription factors can stabilize each other by binding to the same genomic regions in the absence of direct protein-protein interactions, thus cooperating indirectly by competing with nucleosomes[65,80,81]. However, in this study we show that at least a fraction of the PBX1 and HAND2 transcriptional regulators are part of the same protein complexes in hindlimb buds. Such direct PBX1-HAND2 interactions were also detected in forelimb buds, where *Hand2* and *Pbx1* are co-expressed. However, in contrast to hindlimb buds, the functional relevance of PBX1-HAND2 protein interactions in forelimb development remains to be determined. Indeed, unlike in the hindlimb, the distal forelimb phenotype of *Pbx1/2* mutants exhibits low penetrance, likely due to compensation by *Pbx3* in forelimb buds. Therefore, vast numbers of mutant embryos would be required for forelimb-specific genetic analysis, which is beyond the scope of this study.

Several studies have attributed 'pioneer factor' functions to PBX proteins[82–85] during development[86,87]. For example, during zebrafish early embryogenesis TALE regulators access promoters, facilitating chromatin accessibility of transcriptionally inactive genes and preceding HOX protein's binding to initiate transcription[82]. PBX1 binds the promoter/proximal enhancer of doublecortin (*Dcx*) in murine neural progenitors, when chromatin is still compacted, before *Dcx* expression[84]. Our research generates new genome-wide evidence in support of PBX pioneer functions: (1) PBX1 without HAND2 binds to regions that are significantly more accessible and more enriched with H3K27ac than regions bound by HAND2 alone; (2) there is a significant association between repressed promoter elements marked by H3K27me3 and PBX1 binding without HAND2; and (3) PBX1 binds to largely overlapping DNA regions across multiple different embryonic tissues, including hindlimb buds, MF, and BA2. Notably, sites co-bound by PBX1 and HAND2 in hindlimb buds generally overlap regions interacting with PBX1 in the midface, where *Hand2* is not expressed. These results indicate that PBX1 does not require HAND2 to access these regions, pointing to a potential pioneer factor role of PBX in these tissues. Yet, to unequivocally assign pioneer factor functions it will be critical to establish that "PBX-marked" genes are not already primed for transcriptional activation by pre-existing histone modifications[88,89] prior to PBX binding. PBX1/2 requirements in the hindlimb mesenchyme during limb bud initiation support the notion that these TALE regulators have essential functions in multipotent progenitor cell populations[71]. Also, while in the mouse *Hox* gene expression commences only during gastrulation[90], *Pbx* transcripts are already detected in both mouse and human embryonic stem cells[91,92] and then in the blastocyst and morula onward[93], consistent with putative pioneer factor functions.

Our study shows that PBX TALE proteins[3], characterized by a three-amino-acid loop insertion in the homeodomain, a conserved DNA-binding moiety shared by hundreds of transcription factors[94,95], bind indiscriminately to large numbers of CRMs shared across diverse embryonic tissues. Thus, binding of PBX alone is not sufficient for context-dependent functions, whereas PBX1 binding acquires tissue-specific roles via combinatorial interactions with different cofactors, such as HAND2, which confers limb bud functionality (Fig. 9). Cooperative binding with HAND2 is expected to generate quantitative rather than qualitative (i.e. binding/no binding) differences in the levels of PBX occupancy at genomic sites shared across tissues. Such quantitative changes are a feature of continuous networks, in which transcription factors bind a continuum of functional and non-functional sites and regulatory specificity derives from quantitative differences in the DNA occupancy patterns[96]. A similar model defining how TALE factors can activate target genes in different contexts was proposed for MEIS[38,63]. Given that the duration of transcription factor binding to DNA positively correlates with downstream transcriptional output[97,98], we envisage that increased accessibility reflects higher DNA-binding affinity of PBX, and prolonged residence time on chromatin, in the presence of HAND2.

PBX homeoproteins have long been considered as cofactors for HOX proteins, and heterodimerization with PBX has been proposed as a mechanism by which HOX proteins acquire DNA-binding selectivity and specificity[99–103]. However, it remains difficult to conceive how transcription factors with widespread distribution in the embryo, such as PBX proteins, can confer functional specificity to *Hox* genes, which display domain-restricted expression[47]. Challenging the accepted model, we have reported that PBX transcription factors hierarchically control *Hox* gene expression in limb buds[6] and also function in "*Hox*-less" embryonic domains, such as the developing head[60,61,70] (Fig. 9). Now, using the hindlimb bud as a model system, this study provides new evidence that interaction with select cofactors, such as HAND2,

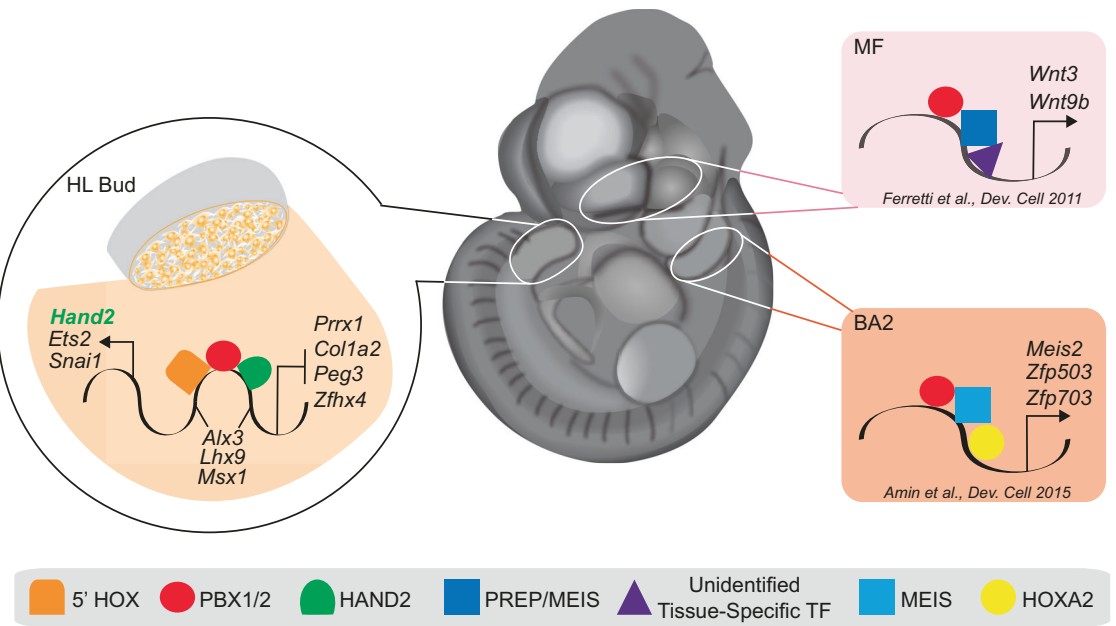

**Fig. 9 | A PBX1/2-HAND2-directed GRN converges on limb transcriptional regulators, as promiscuous PBX DNA-binding acquires tissue-specific developmental functions by interactions with distinct cofactors.** Model depicting how PBX homeoproteins rely on DNA-binding partnerships with different cofactors to attain context-dependent developmental functions. In the midface (MF), a "Hox-less" domain, binding of PBX1/2 together with PREP/MEIS at *cis*-regulatory elements of *Wnt3* and *Wnt9b* (pink-shaded area) directs their expression during upper lip/primary palate fusion[60]. In BA2, PBX1/2 activate a distinct transcriptional program by binding with HOXA2 and MEIS[38] to specific *cis*-regulatory regions at the *Zfp503*

and *Zfp703* loci (ochre-shaded area). In a subset of posterior hindlimb bud (HL) mesenchymal cells (orange shading reflects co-expression of *Pbx1/2*, red, and *Hand2*, green), a PBX1/2-directed GRN acquires early limb functionality via interaction with HAND2. This spatio-temporally constrained GRN (orange-shaded area) positively regulates known limb transcription factors, including *Ets2*, *Snai1* and *Hand2* itself, while it represses others, such as *Prrx1*, *Col1a2*, *Peg3* and *Zfh4*. Lastly, PBX1/2 together with HAND2 also co-regulate other target genes (*Alx3, Lhx9, Msx1*) in a discordant manner.

restrains PBX proteins directing them to execute specific developmental functions in distinct embryonic tissues (Fig. 9). These findings impact our understanding of how promiscuous transcription factors achieve developmental specificity, shedding light on previously unknown mechanisms underlying vertebrate tissue patterning and organogenesis.

## Methods

### Ethics statement and approval of animal research
All animal experiments were performed in accordance with national laws and approved by the local regulatory bodies and authorities as mandated by law in the United States and Switzerland. Experiments with mice and embryos performed at Weill Cornell Medical College and at UCSF were approved by the Institutional Animal Care and Use Committees (IACUC) following their guidelines for housing, husbandry, and welfare. Animal work at Lawrence Berkeley National Laboratory (LBNL) was approved by the LBNL Animal Welfare Committee. Animal studies done in Switzerland at the University of Basel were regulated by animal research permits approved by the Regional Commission on Animal Experimentation and the Basel Cantonal Veterinary Office.

### Generation of mouse embryos
All mutant alleles were previously described and the genotyping was done as previously described for the conditional *Pbx1*[27]*, Pbx2*[10]*, Hand2*[3xFLAG] alleles[24], the constitutive and conditional *Hand2* allele[20] and the *Hoxb6Cre*[30], *Hoxb6CRE-ERT*[31], and *Msx2Cre*[28,29] transgenes. Wild-type (Swiss Webster) mice were purchased from Charles River Laboratories. Female mice older than 6 weeks were put in natural timed mating to obtain embryos at gestational day E10.5 (noon of the day of the plug was determined as E0.5). Mouse embryos were collected from pregnant females after euthanasia and confirmed death.

All mouse embryonic stages are indicated in figures and/or figure legends. Due to genetic complexity, mice and embryos had to be genotyped prior to analysis with exception of the *LacZ* reporter assays. All analyses included embryos of both sexes, i.e. did not discriminate between male and female embryos, given that the phenotypes observed are fully penetrant in all animals.

### Statistics and data reproducibility
No statistical methods were used to predetermine sample sizes, but all sample sizes are based on benchmarked standards in the field. Except for rare, clear technical failures no data were excluded from the analyses. All omics-datasets were collected following the ENCODE guidelines stating that experiments should be performed with minimally two biological replicates. For ATACseq and bulk RNAseq, 3 biological replicates were analyzed, given that these procedures require smaller numbers of embryos. ChIPseq was performed using pools of 60–80 embryonic hindlimb buds per replicate. The high quality and reproducibility of both replicates resulted in statistical significance of the peaks called. To minimize batch-to-batch variation for scRNAseq, the dataset at E10.5 was collected using 10 wild-type pooled embryos. For skeletal preparations, whole-mount RNA in situ hybridization, RNAScope, immunofluorescence and *LacZ* reporter assays in transgenic founder embryos, minimally 3 independent biological replicates were analyzed per genotype and developmental stage. Embryos were isolated from different females and analyzed in two completely independent experiments at minimum. The number of embryos analyzed and founder embryos for each *LacZ* transgenic reporter assay are indicated in all figure legends and tables.

### Skeletal preparations
Due to embryonic lethality of the mouse mutants under analysis, limb skeletal elements were analyzed at embryonic day (E) 14.5. Embryos

were stained for cartilage and bones using standard Alcian blue and Alizarin red staining to obtain skeletal preparations, as described[6,8,20]. Specifically, embryos were washed in hot water at 65 °C to facilitate tissue maceration and permeabilization and removal of skin, eyes, internal organs, and adipose tissue. Following fixation in 95% EtOH O/N at room temperature, embryos were placed in acetone again O/N at room temperature. Subsequently, embryos were stained for cartilage by submerging them in an Alcian blue solution as described[6,8,20] and incubated O/N at room temperature. Embryos were then de-stained by washing them in two changes of 70% EtOH prior to incubation in 95% EtOH O/N. To pre-clear tissues, 95% EtOH was removed and a 1% KOH solution was added for 90 min at room temperature. Once the KOH solution was removed, it was replaced with Alizarin red solution for 3–4 hrs at room temperature, as described[6,8,20]. The Alizarin red solution was then replaced with glycerol:1% KOH (20%:80%) O/N at room temperature. The specimen were then cleared and the excess red color removed by placing them in 1% KOH solutions of decreasing strengths at room temperature, as follows: 80%:20%, 60%:40%, 40%:60%,and 20%:80% of 1% KOH:glycerol. Samples were transferred in 100% glycerol for long-term storage.

## Whole-mount RNA In Situ Hybridization

Whole-mount in situ hybridization was performed as described[5,6,20]. In brief, embryos were rehydrated and pretreated with Proteinase K, and then hybridized O/N at 70 °C with either sense or antisense riboprobes at a final concentration of 1 μg/ml in incubation buffer containing 50% formamide, 5× SSC, 50 μg/ml yeast RNA, 1% SDS, 50 μg/ml heparin, and 0.1% CHAPS detergent (ThermoFisher Scientific). In situ hybridization probes were those used by Capellini et al.[6]. Embryos were then washed through a series of SSC solutions (5× SSC and 2× SSC, three times each for 30 min, and one time each in 0.2× SSC and 0.1× SSC for 30 min, respectively) at 70 °C. After a brief rinse in Tris-buffered saline/0.1% Tween (TBST), embryos were incubated in 10% blocking reagent (ThermoFisher Scientific, # R37620) as described[5,6,20] and the positive signals were detected by AP-conjugated anti-digoxigenin antibody (Roche Diagnostics GmbH, Mannheim, Germany) at 1:5000 dilution. Following washing in TBST, embryos were incubated in NBT/BCIP in NTMT buffer (Roche Diagnostics GmbH, Mannheim, Germany) following the manufacturer instructions, until color fully developed. Positive hybridization was visualized by purple (NBT/BCIP) signal. At least 3 embryos for each genotype and developmental stage were analyzed to establish reproducibility.

## Immunofluorescence

Embryos were harvested, fixed O/N at 4 °C in 4% PFA in PBS, rinsed in PBS, cryoprotected in 30% sucrose O/N at 4 °C, then embedded in OCT compound and cryosectioned at 12 μm per section. Slides were blocked for 1hr with 10% fetal bovine serum (FBS)/PBS and incubated O/N in 0.1% bovine serum albumin (BSA) with primary antibodies (Ab). Primary Abs were: PBX1 (Cell Signalling, #4342, 1:200); FLAG M2 (Sigma, F1804, 1:500). Primary Ab binding was detected by AlexaFluor-conjugated secondary Abs (Invitrogen) at 1:1000 dilution. Specifically, a donkey anti-rabbit IgG (H + L) highly cross-adsorbed secondary Ab (AlexaFluor™ 647; #A-31573) was used to detect PBX1, and a donkey anti-mouse IgG (H + L) highly cross-adsorbed secondary Ab (Alexa Fluor™ 488; #A-21202) was used to detect FLAG M2. Nuclei were stained with DAPI (Sigma). Fluorescence imaging was performed using a Leica SP5 confocal microscope.

## Single-cell RNAseq

Dissected hindlimb buds were collected from 10 embryos at E10.5 (37-40 somites) in cold PBS.

Tissues were dissociated to single cells using an enzymatic cocktail of Liberase and DNase I for 10 min at 37 degrees. Cells were passed through a 45 μm strainer to remove aggregates from the single-cell suspension. Dead cells (≤20%) were eliminated with a 'Dead Cell Removal Kit' with magnetic beads (MACS, Milteny Biotech). Live cells from pooled hindlimb buds were loaded into one well for single-cell capture using the Chromium Single-Cell 3′ Reagent Kit V2 (10X Genomics). Libraries were prepared using the Chromium Single-Cell 3′ Reagent Kit V2, and each sample was barcoded with a unique i7 index. Libraries were pooled and sequenced using an Illumina NovaSeq sequencer. The Cell Ranger v2.2.0 pipeline from 10X Genomics was used for initial processing of raw sequencing reads. Briefly, raw sequencing reads were demultiplexed, aligned to the mouse genome (mm10), filtered for quality using default parameters, and UMI counts were calculated for each gene per cell. Filtered gene-barcode matrices were then analyzed using Seurat v4.1 R package[104]. Cells were filtered to ensure inclusion of only those showing a number of total expressed transcripts between 3000 and 25,000, corresponding to at least 1000 expressed genes, with the mass of transcripts derived from the mitochondrial chromosomes representing less than 10% of the total. Data were normalized using scTransform[105], using the best 5000 features. Cell clusters were identified by constructing a shared nearest neighbor graph followed by a modularity optimization-based clustering algorithm (Leiden algorithm[106]) using the top 60 principal components as determined by PCA. Clustering was performed at multiple resolutions between 0.2 and 2, and optimal resolution was determined empirically based on the expression of known population markers (resolution = 0.8). Cells were visualized in two-dimensional space using Uniform Manifold Approximation and Projection (UMAP) dimensional reduction. Markers for each cluster were identified using the FindAllMarkers function using the Wilcoxon test and setting the *min.pct* to 0.1 and *logfc.threshold* to 0.25. Cluster identity was manually annotated based on the expression of known marker genes. To determine how well each target gene was co-expressed with either *Pbx1/2*, *Hand2*, or both, we focused on the mesenchymal clusters and used as a statistical threshold an 'area under the curve' (AUC) ≥0.55. Mesenchymal cells were re-clustered testing resolution values between 0.2 and 2.0 (using a step of 0.1), and the individual solutions were evaluated using the silhouette (clusterCrit R package). The solution providing the highest value of silhouette was retained. The resulting clusters were automatically annotated using scType[107], and marker genes from distinct mesenchymal cells subpopulations as identified by a recent study[37]. Dorothea[57] was used to retrieve the computationally predicted target genes of PBX1/2 and HAND2. Then these lists of genes were used to score each single-cell for their overall expression, using the function AddModuleScore from Seurat v4.1.

## Bulk RNAseq

Mutant and wild-type hindlimb buds from littermate embryos were dissected individually and snap-frozen in dry ice or liquid nitrogen. After genotyping, each biological replicate consisted of a pair of hindlimb buds isolated from one mutant or wild-type embryo. A total of 3 biological replicates were analyzed per genotype. RNA was extracted using the RNeasy Plus Micro kit (Qiagen, #74034) and total RNA quantified using the Qubit RNA HS Assay Kit (Invitrogen, #Q32852). RNA quality was determined with the RNA 6000 Pico kit (Agilent, #5067-1513) on a 2100 Bioanalyzer (Agilent), or the Fragment Analyzer (Advanced Analytical) High Sensitivity RNA kit. All RNA samples for library preparation had a RIN>9. PolyA RNAs were captured with the NEBNext Poly(A) mRNA Magnetic Isolation Module (NEB, #E7490). RNA sequencing libraries were prepared from 100 ng of RNA using the non-directional kit NEBNext Ultra™ II RNA Library Prep Kit for Illumina (NEB, #E7775). Library size and quality were checked using an Agilent 2100 Bioanalyzer with the High Sensitivity DNA kit (Agilent, #5067-4626), or the Fragment Analyzer CRISPR discovery kit. Library DNA concentration was determined with the QuBit dsDNA HS Assay kit (Invitrogen, #Q32854). *Pbx1/2* libraries were sequenced with the Illumina HiSeq 4000 to generate 50 base pair (bp) single-end reads. *Hand2*

libraries were sequenced using NextSeq500 to generate 75 bp single-end reads. Reads for each tissue were mapped against the mouse genome (mm10) using the Tophat 2 aligner (version 2.0.13) with default parameter settings except for setting the flag --no-coverage-search. Expression levels for each tissue were initially quantified using htseq-count. Differential expression analyses (*Pbx1$^{f/f}$;Pbx2$^{-/-}$;Hoxb6$^{Cre/+}$* versus littermate controls; and *Hand2$^{f/f}$;Hoxb6$^{Cre/+}$* versus littermate controls) was performed using the Bioconductor package edgeR[108]. Briefly, after estimating global and gene-wise dispersion parameters, normalization was performed using TMM (trimmed mean of M-values)[109]. Genes with a fold change ≥1.2 or ≤−1.2 and a FDR ≤ 0.05 were defined as differentially expressed genes (DEGs).

## ChIPseq

Embryonic hindlimb buds and embryonic midfaces (the latter dissected as described[61]) were isolated from wild-type Swiss Webster mouse embryos (E10.5 and E11.5), and immediately crosslinked for 10 min in 1% formaldehyde (Electron Microscopy Sciences, #15710). ChIP assays were performed as previously reported[38,39,61]. The crosslinked material was sonicated to 200–500 bp DNA fragments with a Diagenode Bioruptor or Covaris S220 sonicator. ChIPseq was performed pooling 60–80 embryonic hindlimbs per replicate. Crosslinked and sonicated extracts were incubated with specific antibodies (5 µg) O/N at 4 °C followed by 30 min incubation with Dynabeads protein A (PBX1, H3K27ac and H3K27me3 ChIPseq) or 60 min incubation with Dynabeads protein G (HAND2$^{3XF}$ ChIPseq) to immunoprecipitate (IP) the specific chromatin complexes. IP and input DNA were purified using the MicroChIP DiaPure kit (Diagenode, #C03040001). The following Abs used were for IP: PBX1 (Cell Signaling, #4243S); FLAG for HAND2 (Sigma-Aldrich, F1804); H3K27ac (Abcam, ab4729); and H3K27me3 (Millipore, #07-449). Each ChIP assay included two independent biological replicates. Following ChIP, DNA libraries were constructed using the MicroPlex Library Preparation Kit v2 (Diagenode, C05010012) and sequenced using an Illumina HiSeq 4000 to generate 50 bp single-end reads. ChIPseq reads were aligned to the mm10 release of the mouse genome (Dec. 2011, GRCm38) using Bowtie[110] with parameters -v 2 -m 1. Peak calling was performed using Model-Based Analysis for ChIPseq (MACS) v1.4[111] with matched input DNA as control and parameters --gsize = mm --bw = 150 --nomodel --shiftsize = 100. Each experiment was performed in duplicate; for HAND2 and PBX1 ChIPseq analysis, peaks detected in both replicates were merged using a statistical method that takes into account the combined statistical evidence from the two replicates[112] (MSPC; parameters: -r biological -s 1E-10 -W 1E-6). Genome-wide, scaled wiggle tracks were converted to bigwig using wigToBigWig. Then ChIPseq peaks were associated to genes using custom scripts. Hypergeometric Optimization of Motif EnRichment (HOMER[42]; v4.9) was used to perform enrichment analysis for known transcription-factor-binding sites as well as de novo motif discovery.

## ATACseq

We used the Assay for Transposase-Accessible Chromatin (ATACseq) protocol, as described[40] with minor modifications. About 75,000 cells from a pair of mouse hindlimb buds were used. Three biological replicates were analyzed. Peak calling was performed with MACS v1.4, using the following parameters: --gsize = mm --bw = 150 --nomodel --nolambda --shiftsize = 75. Called peaks from the 3 replicates were combined using the same approach described for ChIPseq analysis.

## Co-immunoprecipitation and Western blot

Mouse limb buds isolated at E10.5 were lysed in lysis buffer (Tris-HCl 50 mM pH 7.4, NaCl 150 mM, MgCl2 2.5 mM, Triton X-100 1%, Sodium Deoxycholate 0.2%, EDTA 1 mM, Glycerol 10%) supplemented with protease and phosphatase inhibitors (Sigma-Aldrich). For HAND2 immunoprecipitation, E10.5 limb buds were collected from

*Hand2$^{3xFlag/3xFlag}$* embryos carrying a 3xFLAG epitope tag into the endogenous HAND2 protein[24]. Wild-type limb buds were used as negative controls. Flag-tagged HAND2 was pulled down by O/N incubation of 400 µg total protein lysates with DYKDDDDK Fab-Trap Agarose beads (ChromoTek, #ffak-20) at 4 °C. For endogenous PBX1 immunoprecipitation, E10.5 limb buds were collected from wild-type (Swiss Webster) mice. One thousand eight hundred micrograms of total protein lysate were pre-cleared with Protein A-Dynabeads (ThermoFisher Scientific, #10001D) for 2 hrs at 4 °C followed by an O/N incubation with 5 µg of polyclonal anti-PBX1 Ab (Cell Signaling #4342S) at 4 °C. For controls, 5 µg of normal rabbit IgG (R&D, #AB-105-C) was used for immunoprecipitation. The next day, protein A-Dynabeads were added to the antibody-lysate mix and incubation continued for additional 2 hrs at 4 °C. DYKDDDDK Fab-Trap Agarose beads or Protein A-Dynabeads were washed in lysis buffer without Sodium Deoxycholate and resuspended in denaturing buffer (Thermo-Scientific, #NP007) with β-mercaptoethanol. Samples were boiled for 5 min and loaded into 12% or 13.5% SDS–PAGE gels for Western blots. Proteins were transferred onto a PVDF membrane (Millipore-Sigma #IPVH00010) at 100 V for 140 min at 4 °C. Membranes were blocked with 5% nonfat milk (BioRad, #1706404) in TBST-T (Tris 10 mM, NaCl 150 mM, Tween-20 0.05%) and subsequently incubated O/N at 4 °C with Abs against HAND2 (Santa Cruz Biotechnology A-12, #sc-398167, 1:1000 dilution), PBX1 (Cell Signaling, #4243S, 1:5000 dilution) and FLAG (Millipore-Sigma clone M2, # F1804 1:2000 dilution) in TBST. Secondary Abs used for detection were HRP-conjugated goat anti-mouse (H+L) IgG (Proteintech, #SA00001-1, 1:5000 dilution), HRP-conjugated goat anti-rabbit IgG (BioRad, #1706515, 1:5000 dilution), and TidyBlot Western Blot Detection Reagent (BioRad #STAR209P, 1:200 dilution, for immunoprecipitated PBX1 only). Abs bound to target proteins were detected with West Dura ECL substrate (ThermoFisher Scientific, #34075).

## RNAscope

*Pbx* and *Hand2* mutant and littermate control embryos were assayed by RNAScope as described[58] with the following modifications: *Pbx1cKO$^{Mes}$;Pbx2$^{-/-}$* and *Hand2cKO$^{Mes}$* mutant and control littermate embryos were collected and fixed with 4% paraformaldehyde in 1× PBS O/N at 4 °C. Embryos were then immersed in 30% sucrose, and embedded in Epredia™ Neg-50™ Frozen Section Medium. Frozen blocks were cut to 14-µm-thick cryosections that were air dried at −20 °C for 1 h and stored at −80 °C. Slides carrying hindlimb bud sections were thawed and washed with 1X PBS to remove excess freezing medium before use. Slides were assayed using an RNAscope™ Multiplex Fluorescent Reagent Kit V2 following the manufacturer's instructions with some modifications. By skipping target retrieval steps and treating slides with Protease Plus for no longer than 10 min, damage to fragile embryonic tissues was avoided. Probe mixes were hybridized for 2 h at 40 °C in a HybEZ™ II Oven (Advanced Cell Diagnostics, Newark, CA). The following probes were used: *Pbx1* (ACD, #435171); *Prrx1* (ACD, #485231-C2); *Snai1* (ACD, #451211-C2), and *Hand2* (ACD, #49821-C3). The appropriate HRP channels were developed with Opal™ 520, TSA™ Cy3 Plus, and Cy5 Plus (PerkinElmer) dyes. Following DAPI staining and mounting with ProLong™ Gold (Invitrogen), sections were imaged using a Zeiss Axio Observer.Z1 with a Plan-Apochromat 40×/1.3 Oil DiC (UV) VIS-IR M27 immersion objective. Tile-scan images were acquired, stitched, and adjusted using Zen2™ (Zeiss).

## *Hand2* candidate enhancer identification for analysis of *LacZ* activity

We intersected our ChIPseq and ATACseq datasets from E10.5 hindlimb buds to identify all regulatory elements within the *Hand2* TAD[46]. We selected all PBX1-bound called in both replicates that show at least 15-fold-enrichment, excluding *Hand2* promoter regions. To identify

the candidate enhancers that control *Hand2* expression in the limb bud mesenchyme, we intersected the evolutionarily conserved non-coding elements with the PBX1-bound regions that were also enriched for H3K27ac marks and located in accessible chromatin (ATAC-seq peaks).

### Transgenic mouse reporter assays

Transgenic mouse assays were performed using the *Mus musculus* FVB strain. Embryos of both sexes were analyzed. Sample size selection and randomization strategies were conducted as follows: sample sizes were established empirically based on previous experience in transgenic mouse assays for >3000 total putative enhancers (VISTA Enhancer Browser: https://enhancer.lbl.gov/). Mouse embryos were excluded from further analysis if they did not carry the reporter transgene or if the developmental stage was not correct. All transgenic mice were treated in identical experimental conditions. Randomization and experimenter blinding were unnecessary and not performed. For validation of in vivo limb enhancer activities, conventional transgenic mouse *LacZ* reporter assays involving an hsp68 minimal promoter (*Hsp68-LacZ*) were performed as described[113,114]. For comparison of wild-type and mutagenized enhancer versions (mm1689), enSERT was used for site-directed insertion of transgenic constructs at the H11 safe-harbor locus[48]. By this approach, Cas9 and sgRNAs were co-injected into the pronucleus of FVB single-cell-stage mouse embryos (E0.5) together with the targeting vector encoding the candidate enhancer element upstream of the *Shh*-promoter-*LacZ* reporter cassette (*Shh-LacZ*)[48]. The relevant genomic coordinates of the tested enhancers are listed in Supplemental Table 2. The predicted enhancer elements were PCR-amplified from mouse genomic DNA (Clontech) and cloned into the respective *LacZ* expression vector[114]. For enSERT, embryos were excluded from further analysis if they did not carry the reporter transgene in tandem. Pseudo-pregnant CD-1 recipient females were used for embryo transfer. Transgenic embryos were collected at E10.5 or E11.5 and stained with X-gal using standard techniques[114].

### Mouse *Hand2* enhancer mutagenesis

The coordinates of enhancer mm1689 are shown in Supplementary Table 2. Binding motifs for PBX and HAND were disrupted following the same strategy. In all cases, the binding sites were disrupted by substituting T/A bases into C and C/G bases into A. To allow direct comparison, the same number of bases were altered in both mutagenesis experiments. The binding motifs that were mutagenized follow: Enhancer mm1689 encodes 7 PBX or PBX-HOX binding sites at these base-pair positions: 703, 1059, 1366, 1482, 1560, 1765 and 1902. Enhancer mm1689 encodes 8 HAND complete or partial binding sites. The bases disrupted were the 6 core bases of the HAND motif at these base-pair positions: 217, 959, 1289, 1376, 1633, 1680, 1701 and 1742.

### Reporting summary

Further information on research design is available in the Nature Portfolio Reporting Summary linked to this article.

## Data availability

ChIPseq, ATACseq, scRNAseq and RNAseq datasets have been deposited in the NCBI GEO database under the identifier GSE197859. Pre-processed scRNAseq data have also been deposited in Zenodo [https://zenodo.org/record/7884496#.ZFA4dexBzvU]. Source data are provided with this paper.

## Code availability

The R code to reproduce the scRNA-seq data analyses has been deposited in Zenodo: https://zenodo.org/record/7884496#.ZFA4dexBzvU. Additional data supporting the reported findings are available upon request.

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

## Acknowledgements

We are grateful to Mr. R. Aho for all artwork; Drs. D. Penkov and F. Blasi for ChIPseq protocols and hospitality in their laboratory; A. Baur and O. Romashkina for expert technical assistance; A. Offinger and the team for mouse care; M. Kmita and I. Desanlis for sharing datasets; Selleri laboratory and Developmental Genetics members, as well as Drs. Panagiotakos, Bush, Blelloch, Wagner, and Lacy for useful discussions. The research was funded by the National Institutes of Health (grants R01HD043997 and R01DE028324) and by the University of California, San Francisco Chancellor's Funds to L.S.; by the Swiss National Science Foundation grant no. 310030_184734 and core funding from the University of Basel to R.Z. and A.Z.; the Swiss National Science Foundation grant no PCEFP3_186993 to M.O. and the National Institutes of Health grant R01DE028599, UM1HG009421 and R01HG003988 to A.V. Research at E.O. Lawrence Berkeley National Laboratory was performed under US Department of Energy contract DE-AC02-05CH11231, University of California. M.L. was the recipient of a postdoctoral fellowship from the American Association for Anatomy and I.B. was funded by an Imperial College Research Fellowship and by the Medical University of Vienna. Fruitful discussions and interactions initiated this project while R.Z. and subsequently N.B. were sabbatical visitors with L.S. at UCSF.

## Author contributions

L.S. and R.Z. designed and supervised the study; A.Z. designed and supervised the *Pbx-Hand2* genetic interaction experiment; M.L. conducted most multi-omics and WISH experiments; I.B., P.Z., and N.B. performed the bioinformatic analyses; M.O. and D.D. directed all transgenic mouse reporter assays and analyses; V.H.A. optimized and conducted PBX-HAND2 Co-IP experiments; A.M. performed the *Pbx-Hand2* genetic interaction experiment and generated all *Hand2* embryos; B.C. generated *Pbx1/2* mutant embryos and performed RNAScope for gene target validations; A.G. conducted *Hand2* multi-omics and IF experiments; J.D.B. generated *Pbx1/2* mutant embryos for skeletal analysis and IF experiments; J.Z. and S.M. provided the *Hoxb6CreERT* mouse line; T.C. conducted the first tissue-specific *Pbx1/2* inactivation; and R.Z. manually annotated and curated the PBX-HAND2 GRN. M.L., I.B., and L.S. wrote the manuscript, which was critically reviewed by R.Z. N.B., M.O., A.Z., T.C., and A.V. gave important input on the manuscript, which was read and approved by all co-authors prior to submission. The final editing of the manuscript for acceptance was done by L.S., I.B., M.L., A.Z., and R.Z.

## Competing interests

The authors declare no competing interests.

## Additional information

[1]Program in Craniofacial Biology, Institute for Human Genetics, Eli and Edythe Broad Center of Regeneration Medicine and Stem Cell Research, Department of Orofacial Sciences and Department of Anatomy, University of California San Francisco, San Francisco, CA, USA. [2]Center for Cancer Research, Medical University of Vienna, Vienna, Austria. [3]Environmental Genomics and Systems Biology Division, Lawrence Berkeley National Laboratory, Berkeley, CA, USA. [4]Department for Biomedical Research (DBMR), University of Bern, Bern, Switzerland. [5]Department of Cardiology, Bern University Hospital, Bern, Switzerland. [6]Developmental Genetics, Department Biomedicine, University of Basel, Basel, Switzerland. [7]School of Medical Sciences, University of Manchester, Manchester, UK. [8]Cancer and Developmental Biology Laboratory, Center for Cancer Research, National Cancer Institute, Frederick, MD, USA. [9]Department of Human Evolutionary Biology, Harvard University, Cambridge, MA, USA. [10]Broad Institute of MIT and Harvard, Cambridge, MA, USA. [11]US Department of Energy Joint Genome Institute, Lawrence Berkeley National Laboratory, Berkeley, CA 94720, USA. [12]School of Natural Sciences, University of California, Merced, Merced, CA 95343, USA. [13]These authors contributed equally: Marta Losa, Iros Barozzi. [14]These authors jointly supervised this work: Rolf Zeller, Licia Selleri. ✉e-mail: licia.selleri@ucsf.edu

