## [Peer Review File · Nature Communications]

A spatio-temporally constrained gene regulatory network directed by PBX1/2 acquires limb patterning specificity via HAND2Editorial Note: Parts of this Peer Review File have been redacted as indicated to remove third-party material where no permission to publish could be obtained.

REVIEWER COMMENTS

Reviewer #1 (Remarks to the Author):

In this elegant and interesting study, Losa, Barozzi et al. address a very important question, namely how PBX TALE transcription factors, generally rather ubiquitously expressed, achieve functional specificity in a tissue- and/or structure-specific manner during *in vivo* embryo development and morphogenesis. As a suitable model system, the authors investigate PBX1/2 function during hindlimb development in the mouse. Using a comprehensive and impressive combination of conditional tissue-specific and time-controlled knockouts, single cell and bulk RNAseq, chromatin profiling of transcription factor binding sites, *in vivo* mutagenesis of cis-regulatory sequences, and mutant phenotype comparisons, the authors show that PBX1/2 and HAND2 genetically interact and are collaboratively required in a subset of posterior-proximal hindlimb mesenchymal cells. They further show that PBX1 directly binds and regulate Hand2 expression through hindlimb-specific cis-regulatory modules. Most importantly, they also reveal that HAND2 selects and binds a subset of PBX1/2-bound regulatory sites to drive a gene regulatory program for proximo-distal hindlimb and digit patterning.

The presented results are convincing, nicely documented, and the paper is well written. The conclusions are compelling, clear, and well supported by the presented data. These results provide important and novel insights into how promiscuous transcription factors achieve tissue and/or cell-type developmental specificity, as well as how the early limb program is set in place by key transcription factors.

Specific comments:

The study would definitely benefit from gathering additional mechanistic information about PBX/HAND2 cooperation on target genes. For instance, do PBX/HAND2 directly interact? Even if these factors do not directly interact, is pre-binding of PBX1/2 necessary for HAND2 selection of cis-regulatory elements? Do PBX1/2 function as pioneer factors on chromatin? Ideally, suitable experiments to try and address some of these questions could be testing accessibility of PBX1/2 binding sites in the PBX1/2 knockout by ATACseq, and/or HAND2 binding by ChIPseq or ChIP-PCR at selected sites in PBX1/2 knockout limb buds. However, I am aware that such experiments could be technically very challenging *in vivo* and possibly unfeasible, given that Pbx1/2 and Hand2 are only co-expressed in a small subpopulation of mesenchymal hindlimb cells, as shown by single cell RNAseq. However, the authors could at least show a co-immunoprecipitation in wild type limb buds to assess whether PBX and HAND2 can interact at the molecular level.

Reviewer #2 (Remarks to the Author):

The paper by Losa and collaborators presents an analysis of hindlimb development regulation by Pbx1 and Pbx2 as well as an analysis of Pbx1's interaction with Hand2 in this process. The authors show that Pbx1/2 genes are required in the limb mesenchyme for the formation of the hindlimb. They also show that the morphological and molecular phenotypes observed in Pbx1/2 mutants are similar to those observed in Hand2 mutants. The paper presents a large amount of multi-omic analyses focusing on Pbx1 and Hand2 genome binding. The authors suggest that Pbx binds promiscuously to many genomic regions in different tissues where it acts as a pioneer factor, and that the tissue specificity is achieved by Pbx's cooperation with distinct tissue-specific transcriptional regulators. In this respect they consider the differential binding of Pbx to targets in the hindlimb, the midface and the second branchial arch, paying attention to the presence or absence in the neighbouring regions of possible tissue specific collaborators like Hand2 or Hox2. The manuscript is clearly written and the experiments

carefully done, however many of the conclusions are based on correlations suggested by the multi-omic data which are not independently confirmed by direct biochemical or genetic experiments. Thus, although the proposed hypothesis is credible, the authors do not present experiments directly supporting it.

I find there may be a major problem with the results. The authors present careful experimental analysis with tamoxifen to find the temporal developmental window when the Pbx1/2 proteins are required for limb development. The conclusion is that hindlimbs develop normally when tamoxifen is administered at or after stage E10, suggesting Pbx function is required earlier. Despite this, all multi-omic analyses are performed using limbs at E10.5.

Another issue is that, even if we accept the multi-omic assays performed at those later stages are informative, the multi-omic data should have been experimentally confirmed on some of the targets.

Thus, when in line 404 the authors claim that their "results demonstrate that promiscuous PBX transcription factors occupy a vast pool of shared genomic regions during embryonic development, but acquire restrained tissue specificity via cooperation with distinct transcriptional regulators, such as HAND2 in the developing hindlimb.", they are proposing a believable hypothesis that has not been demonstrated.

Most of the data presented in the manuscript are sophisticated correlations that require direct molecular or biochemical confirmation by analysing any of the direct target genes detected in the study.

In the manuscript (most clearly in Fig. 8) there is an implicit suggestion that Hand2 autoregulates its expression in combination with Pbx. If this was true, the enhancers found in the Hand2 regulatory region could provide the perfect system to prove the hypothesis. By mutating putative Pbx-binding sites, the authors have already gone halfway into it, providing compelling evidence that one of Hand2 CRMs (mm1689) is directly regulated by Pbx1. It would have been crucial to test if this CRM is also bound by Hand2 and if the enhancer's expression requires Hand2 function in a feedback loop. If this was the case, such an element would serve to test if Hand2 increases Pbx binding to the CRM or if it collaborates as a tissue specific factor inducing localized transcription. However, in the current experiments the authors only analyse the enhancer for Pbx interactions without providing experiments to prove its possible regulation by Hand2 (neither EMSA nor directed DNA site mutagenesis has been done).

The authors genetically demonstrate that Hand2 and Pbx1 genes are collaborating during hindlimb development, however it is not clear if this is due just to Pbx1 regulating Hand2 expression, or due to direct collaborative interactions between Pbx and Hand2 proteins when activating the CRM of their direct targets.

Minor issues:

Some comments could be toned down. For example, in line 489 of discussion:

"Our de novo motif analysis provides evidence that PBX1 can bind DNA without HAND2, but likely together with other TALE factors such as MEIS1/PREP1 in hindlimb buds..."

It is true that this is suggested by the authors' results, but the fact that Pbx binds DNA with TALE factors in regions that do not express Hand2 is not too surprising.

Line 512 of discussion (point 3): I find unclear what this means

Some references are incomplete. For example, Refs 38, 47, 59 and 69.

Reviewer #3 (Remarks to the Author):

This MS by Losa et al. reports on the comprehensive study of gene control by PBX1/2 and HAND2 transcription factors for early hindlimb patterning in the mouse. The question is approached through classical and conditional knockdown technologies, allowing for spatial and temporal restricted loss of function, and a combination of omics approaches, including single cell and bulk RNA-seq, as well as genomic profiling for PBX1, HAND2 and histone marks, associated to ATAC-seq chromatin accessibility assays. The work is technically sound, with an elegant combination of classical and modern molecular genetics, gene expression and genomic profiling. The data is solid, clearly presented, and allows for the emergence of a detailed molecular view of the events controlling hindlimb development, which has served as an important paradigm in developmental biology. The work supports functional cooperativity of PBX1/2 and HAND2, and identifies downstream components constitutive of the GRN at work during early hindlimb patterning. The work also identifies the molecular support for a feed-forward loop whereby PBX1/2 and control the expression of HAND2.

Beyond its contribution to hindlimb development understanding, the work also impacts our understanding on how transcription factors reach specificity. This question is central to transcription factor broadly/ubiquitously expressed but that act in a spatial/temporal restricted manner. This work shows that the broadly expressed PBX1/2 proteins reach specificity in the hindlimb through functional cooperativity with HAND2, expressed in a more restricted manner within the posteriorly located hindlimb mesenchymal cells. Genetics, transcriptomics and genomics data indicate that PBX1/2 functional cooperativity relies on convergent binding to genomic regions, which resident genes are co-regulated by the PBX1/2 and HAND2.

Overall the work exploits hind limb patterning not only to clarify aspects of its morphogenesis, but also to gain insights into the mode of action of PBX proteins. The approaches taken elegantly combine modern molecular mouse genetics and omics approaches to reach solid molecular conclusions. Surprisingly, little is known on GRNs driving developmental processes and on mechanisms tuning such GRNs to different outputs. This comprehensive work in this particular context does provide significant advances that in my opinion merits publication in Nature Communications.

Below, I highlight a number of suggestions that the authors may consider to improve the manuscript.

1- The authors contextualize their work by introducing the HOX-PBX specificity paradigm. This may be rephrased as most of the data generated in this study regards PBX-HAND2 cooperativity. The end of the manuscript does compare PBX-HAND2 and PBX-HOX in the branchial arches, but this is not the core of the manuscript. I would suggest contextualizing the work as a mean to address how broadly expressed transcription factors reach specificity, which in my opinion would better suit the core of the manuscript.

2- Functional cooperativity of PBX and HAND2 is key to the developmental system under study. When discussing the molecular bases for this cooperativity the authors consider a single scenario, that of the facilitation of HAND2 binding by PBX mediated nucleosome clearance. This would imply a temporal sequence of PBX and HAND2 binding. Is there any indication for that? It may also be worth discussing other scenarios: do HAND2 and PBX proteins directly interact? This could be discussed based on the existence/absence of relevant data concerning the role of PBX and HAND2 in other described contexts.

3- The reporting of a GRN mediating PBX1/2 and HAND2 is key to this work. The data reported clearly allows the identification of components of a hindlimb GRN controlled by PBX1/2 and HAND2. The authors however did not push the analysis toward the reconstruction and representation of this GRN. Data for doing so are in the MS (distinction between direct and indirect PBX and HAND2 targets, common/shared targets). This hindlimb gained knowledge could be merged with protein interaction databases to represent the backbone of a PBX/HAND2 GRN. The reason for not doing so may be cryptic, but if existing, may be discussed by the authors.

4- The authors present the genetic interaction studies between PBX and HAND2 as a functional validation of the hindlimb GRN. These genetic interactions provide strong support for the functional cooperativity of PBX and HAND2, and endow functionality to many molecular observations, which are key to the present work. Yet, in my opinion, it does not per se constitute a functional validation of the newly identified GRN. Such an aim would require probing (or extracting from the existing literature) the role of some of the newly identified components, with respect to regulatory relationship and impact on hindlimb development.

5- The authors may explain the rationale beyond introducing conditional loss of PBX1 in a PBX2 deficient background instead of the reverse if there is one.

6- The authors may explain why sc-RNA seq was not employed for monitoring gene expression changes in PBX and HAND2 "mutant" conditions.

7- To establish the PBX dependency of the newly identified HAND2 CRM, the authors probed the activity of a CRM mutated for the PBX sites. Does the CRM (WT) also lose activity in the PBX1/2 mutant context?

8- How the hindlimb GRN described here may compare to a fore limb GRN (to be described) may shortly be discussed, as it may conceptually further explain how an additional level of refinement (specificity) is achieved.

Reviewer #4 (Remarks to the Author):

In this work, Losa and colleagues address a conundrum in developmental genetic: how specific and precise output signals arise from complex Gene Regulatory Networks (GRN), composed of individually widespread-expressed transcription factors. In particular, the authors describe how the PBX1/2 constitutively-expressed transcription factors interplays with another regulated transcription factor, HAND2, to induce specific enhancer activities and control downstream gene transcription. The authors elegantly combine functional genetic inactivations with multi-omics approaches to precisely reconstruct the phenotypic and molecular pathways at play to control the PBX1/2-HAND2 GRN. Overall, this study is important as it is addressing the question of epistatic relationships between factors within a model GRN. The manuscript is well written, with many cross-supportive evidence and solid techniques (e.g., the temporal requirement of Pbx1/2 in the context of hindlimb development using Msx2Cre or Hoxb6CreERT, enhancers validation by transgenesis, mutagenization of transcription factor binding sites within the mm1689 enhancer, single-cell RNA-seq, ChIP-seq of transcription factors and immunofluorescence). However, this reviewer has a few comments that need to be cleared by the authors and found that the intelligibility of the data presented in the figures varies enormously between the different sections of the manuscript. This let us to conclude that this manuscript needs a substantial effort to streamline and harmonize the results presented here.

Major points

1. The authors are defining Pbx3 as a forelimb-restricted transcription factor to justify the stronger phenotypical impact of the PBX1/PBX2 loss in hindlimbs and for using hindlimbs as a model in the present study. Yet, aside from their own study (di Giacomo 2006), Pbx3 was never described as a forelimb specific marker gene to this reviewer knowledge. In fact, transcriptomic comparisons of fore- and hindlimbs at different developmental stages have shown low but similar expression of Pbx3 in both tissues. Therefore, as for many other transcription factors, stronger hindlimb phenotypical outcome of the inactivation of PBX1/2 might originate from a different mechanism. Therefore, unless the authors provide strong evidence that Pbx3 contribution in forelimbs is the cause of the weaker phenotype (i.e., triple inactivation of PBX1/2/3), this reviewer strongly encourages the authors to tone

down the Pbx3 rationale in their experimental design.

2. The single cell analysis presented in Figure 2 is very difficult to apprehend and appear to lack important information.

a. The limb proximo-distal axis should be highlighted on the UMAP by using expression of known marker genes.

b. We believe that a clustering of mesenchymal cells alone could be performed so to have a more focus analysis on relevant cell types. Such a re-analysis could also be used in Figure 5.

c. The authors show 13 types of mesenchymal cluster in Figure 1D and Figure 5 without defining their identity. It is difficult to understand the rationale behind this high number and the function of each cluster. It is thus critical that the authors minimally name them. This reviewer would also propose to reduce the number of these clusters for an easier interpretation and to make it comparable with other published single cell data in the limb (see also comment "b" above). This would ease the interpretation of the large number of violin plots shown in Fig2 D/E that are presently not corresponding to the nomenclature shown in Fig2 B/C.

3. The co-expression analysis presented in Figure 5 is very interesting and clearly pinpoints the importance of cell autonomous GRNs.

a. Yet, the authors do not mention what genes are regulated in a non cell-autonomous way and impacted in other clusters than the ones co-expressing Pbx1 and Hand2. Maybe some of these genes are making sense in regard to the literature?

b. Also is the Dorothea approach a confirmation of the Fig 5A analysis or does it provide new results on the co-expression of PBX/HAND2 and target genes? For now, it is presented as a new result but somehow appear redundant with the previous analysis.

4. The authors describe H3K27me3 as a repressive enhancer mark. This is true but, to this reviewer knowledge, only in very few specific cases. Therefore, the absence of association between PBX2/HAND2 peaks and H3K27me3 is rather biased, and it is unclear if the authors can conclude something from this analysis on enhancers. This reviewer would suggest removing this analysis from the paper or to only focus on repressed/bivalent promoter regions only.

5. The transcription factor peak analysis is so far difficult to apprehend because of the following points:

a. Despite the authors presenting the binding of PBX1 genome wide as highly similar between different tissues, it appears to this reviewer that there is quite some tissue-specificity, especially looking at Fig. 7A. The authors justify here the PBX1 binding changes between cell type as being similar to the differences between ChIP replicates. Yet the cell-type specificity of PBX1 binding is then mentioned as relevant to HAND2/HOXA2 binding, and could therefore not be only a technical bias of ChIP, but a clear biologically relevant synergistic effect. This apparent contradiction needs to be clarified. Moreover, in the present analysis each peak is validated by replicates.

b. Despite the authors clearly showing that HAND2/HOXA2 do not bind the same regions (Fig 7G), the overlap between their respective binding sites and PBX1 cell-type specific sites is not striking (Fig7 D and F). This might be due to a lack of numbers in the Venn diagram or other statistical analyses, or to a lack of clarity in the author's description of the diagrams (lines 386-389).

c. Finally, the example loci selected do not clearly convey the message found genome wide and developed in the text.

6. Generally, the presentation of the conditional allele approach L130-139 is not straightforward to understand. Ideally, the authors could produce a scheme (i.e., as a supplementary figure), to illustrate, to the general audience, where and when during limb development the different Cre lines delete Pbx1.

Other points

1. It is unclear to this reviewer why the authors systematically refer to GO terms in their analyses. It appears a bit as a circular argument as they work with limb tissue, limb associated gene and then, somehow obviously, find these terms associated to their analyses.
2. The authors need to clarify alleles nomenclature: *Hoxb6Cre/+* and *Hoxb6Cre*, *Hoxb6CreERT/+* and *Hoxb6CreERT* (see L143, L144, L149, L150, L153..).
3. L43: GRN is not defined in the Abstract section.
4. L267: what are "complement" of limb enhancers? Are the authors talking about "repertoire" or "sets" of enhancers?
5. L321: sc RNAseq and L474 scRNASeq: harmonize format
6. L445: "Consistent with essential PBX1/2 functions in mesoderm patterning, we reported that they are required to control Polycomb and Hox gene expression in the paraxial mesoderm.". Can the authors clarify this sentence, i.e. PBX1/2 control Polycomb genes or genes that are repressed by Polycomb, such as HOX.
7. L460: Can the authors clarify why, with their data, they conclude that PBX/HAND2 GRN is initiating hindlimb bud patterning?
8. L476-479: the paragraph conclusion sentence is not understandable.
9. Fig2A: Vignettes/graphical presentations of the experiments "bulk ChIP/ATAC WT" and "bulk RNAseq" appear as dispensable to this reviewer.
10. Fig3F-G: Not enough graphical differences between F and G panel results. A title at the top of F (biological processes) and G (mouse phenotype) would help the reader to better appreciate and discriminate those results. This comment is also valid for Fig3D-E.
11. Fig3I-I'-I'': The visualization of the Both.Intergenic (I') and Both.Promoter (I'') subsets is not easy to comprehend (discrimination of the two boxes - solid line versus dash line borders). Maybe the authors could provide a description in the figure legend?
12. Fig4A: H3K27ac ChIP-seq signal seems overall low in *Hand2* gene desert, also very low on the putative *Hand2* enhancers. Could a small fraction of *Hand2*-expressing cells at this developmental stage explain this result?
13. Fig4E-H: Strong differences in the X-gal staining between mm1689 HL E11.5 (Fig4E) and mm1689 WT (Fig4H). Could this be explained by difference in the transgenesis technique used (random versus enSERT transgenesis)?
14. Fig5A: Bulk RNA-seq: $FC \geq 1.2$, can the authors explain why they take such a low cutoff? $FC \geq 1.5$ is generally used in the literature.
15. Fig5D: color of H3K27ac track is not matching with its name (orange versus dark green).
16. Fig8 The message is very difficult to apprehend from the current model. Maybe having the various tissue depictions on top of one another, to compare which factors binds in each tissue could make it clearer? Moreover, the current depiction is misleading in regard to the "discordant" expressed genes (*Alx3/Msx2* and *Lhx9*). Finally, a reader could also assume from this graphical representation that all the target genes depicted here lies in cis.

Reviewer #1 (Remarks to the Author):

In this elegant and interesting study, Losa, Barozzi et al. address a very important question, namely how PBX TALE transcription factors, generally rather ubiquitously expressed, achieve functional specificity in a tissue- and/or structure-specific manner during *in vivo* embryo development and morphogenesis. As a suitable model system, the authors investigate PBX1/2 function during hindlimb development in the mouse. Using a comprehensive and impressive combination of conditional tissue-specific and time-controlled knockouts, single cell and bulk RNAseq, chromatin profiling of transcription factor binding sites, *in vivo* mutagenesis of cis-regulatory sequences, and mutant phenotype comparisons, the authors show that PBX1/2 and HAND2 genetically interact and are collaboratively required in a subset of posterior-proximal hindlimb mesenchymal cells. They further show that PBX1 directly binds and regulate Hand2 expression through hindlimb-specific cis-regulatory modules. Most importantly, they also reveal that HAND2 selects and binds a subset of PBX1/2-bound regulatory sites to drive a gene regulatory program for proximo-distal hindlimb and digit patterning.

The presented results are convincing, nicely documented, and the paper is well written. The conclusions are compelling, clear, and well supported by the presented data. These results provide important and novel insights into how promiscuous transcription factors achieve tissue and/or cell-type developmental specificity, as well as how the early limb program is set in place by key transcription factors.

Specific comments:

➤ The study would definitely benefit from gathering additional mechanistic information about PBX/HAND2 cooperation on target genes. For instance, do PBX/HAND2 directly interact? Even if these factors do not directly interact, is pre-binding of PBX1/2 necessary for HAND2 selection of cis-regulatory elements? Do PBX1/2 function as pioneer factors on chromatin?

Ideally, suitable experiments to try and address some of these questions could be testing accessibility of PBX1/2 binding sites in the PBX1/2 knockout by ATACseq, and/or HAND2 binding by ChIPseq or ChIP-PCR at selected sites in PBX1/2 knockout limb buds. However, I am aware that such experiments could be technically very challenging *in vivo* and possibly unfeasible, given that Pbx1/2 and Hand2 are only co-expressed in a small subpopulation of mesenchymal hindlimb cells, as shown by single cell RNAseq. However, the authors could at least show a co-immunoprecipitation in wild type limb buds to assess whether PBX and HAND2 can interact at the molecular level.

We thank the Reviewer for their appreciation of our work, for the depth of their comments, and for the important suggestions. To address the Reviewer's concerns, first we focused on the optimization of a co-immunoprecipitation assay from embryonic tissues, to assess potential direct interactions of PBX and HAND2 at the molecular level *in vivo*. We conducted co-immunoprecipitation experiments in two ways: 1) using E10.5 *Hand2*^{3XFlag/3XFlag} hindlimb buds and 2) using E10.5 wildtype hindlimb bud samples. In parallel, we performed co-immunoprecipitation experiments also in forelimb buds. We unequivocally demonstrated that PBX1 interacts *in vivo* with HAND2 not only in hindlimb buds, but also in forelimb buds. We have incorporated these new results in revised Fig. 3I,J. Uncropped immunoblots are also available to the Reviewers within the Supplementary Figures file (Uncropped blots).

The second point raised by the Reviewer required addressing the potential presence of sequential binding of PBX1 and HAND2 to DNA, or, in other words, whether pre-binding of PBX1/2 is necessary for HAND2 selection of cis-regulatory elements. We first attempted to address this point indirectly by assessing whether *Pbx1/2* are expressed in the mouse lateral plate mesoderm (LPM) at an earlier time-point than *Hand2*. Indeed, we previously reported that *Pbx1/2* are already expressed in the LPM by E8.5 (Capellini et al, *Development*, 2006). To evaluate whether *Hand2* expression is already expressed in the LPM at E8.5, or commences only during later developmental stages, we conducted RNAscope with *Pbx1* and *Hand2* probes on E8.5 LPM. We found that transcripts for both genes are co-expressed in the LPM at these very early developmental stages, even prior to hindlimb budding (see figure below). Therefore, *Pbx1/2* expression does not precede *Hand2* expression in the LPM, and the two genes are co-expressed in the same cells that will give

rise to the hindlimb bud (see figure below), making it impossible to address the potential presence of sequential binding of PBX1 and HAND2 to chromatin *in vivo* by this indirect approach.

We have not incorporated this new panel showing co-expression in the LPM in the revised manuscript for lack of space, but we are making it available to the Reviewer and to the Editor.

In light of the inconclusive results described above, we pursued a different approach. We collected a large number of *Hand2* mutant embryos and conducted PBX1 ChIPseq in *Hand2* mutant hindlimbs to evaluate whether PBX1 binding was affected - or not - in the absence of HAND2, so as to establish a putative sequential binding of these factors in hindlimb buds. Likely due to the low amount of mutant hindlimb tissue that could be recovered from E10.5 mutant embryos, the first ChIPseq experiment worked sub-optimally, despite the modifications we implemented to our established protocols. We then conducted two additional PBX1 ChIPseq biological replicate experiments using even larger numbers of *Hand2* mutant hindlimbs. Unfortunately, the number of statistically significant peaks called in the two experiments was drastically different. Indeed, we were able to confidently call only 500 and 2,000 peaks, respectively, for the two replicates, again likely due to insufficient starting material. Thus, no unequivocal conclusions could be drawn from this extensive and costly amount of additional work.

Repeating additional ChIPseq assays would require the generation of even larger numbers of mutant E10.5 hindlimbs, which in turn would require setting up again a massive number of timed-matings for the production of vast cohorts of E10.5 mutant embryos (that comprise multiple engineered alleles). This approach would require modifications to our Mouse Protocols, as we would exceed the total number of mice that have been approved for use to our lab in a given year. Alternatively, we could switch to CUT&RUN, a technique that we are currently adopting in our lab to use substantially less starting material. However, the entire study was based on analyses of ChIPseq data sets. Switching now to CUT&RUN would not enable comparisons with the original ChIPseq data sets that comprise a large part of this study. As this Reviewer stated: “*such experiments are technically very challenging in vivo and possibly not feasible, given that Pbx1/2 and Hand2 are only co-expressed in a small subpopulation of mesenchymal hindlimb cells*”. The Reviewer then added: “*the authors could at least show a co-IP in wild type limb buds to assess whether PBX and HAND2 interact at the molecular level.*” We agree with the Reviewer that the ChIPseq experiments are not feasible in a reasonable time-frame. However, we believe that we have now shown a direct interaction between *PBX1* and *HAND2* rigorously and exhaustively in revised Fig. 3I,J, as requested.

Lastly, regarding the putative sequential binding of PBX1 and HAND2 factors in hindlimb buds, we believe we have some *indirect* evidence that PBX1 binding to chromatin precedes HAND2 binding. The evidence supporting this idea derives from the finding that regions bound by both PBX with HAND2 in the hindlimb largely overlap with regions bound by PBX alone in the midface (where *Hand2* is *not* expressed). These results strongly suggests that PBX does not require HAND2 to access these regions, pointing indeed to a potential pioneer factor role for PBX. Accordingly, we have added a new sentence in the revised Discussion that reads:

“Notably, sites co-bound by PBX1 and HAND2 in hindlimb buds generally overlap regions interacting with PBX1 in the midface, where Hand2 is not expressed. These results indicate that PBX1 does not require HAND2 to access these regions, pointing to a potential pioneer factor role of PBX in these tissues.”
(page 16 of the revised manuscript).

Reviewer #2 (Remarks to the Author):

The paper by Losa and collaborators presents an analysis of hindlimb development regulation by Pbx1 and Pbx2 as well as an analysis of Pbx1's interaction with Hand2 in this process. The authors show that Pbx1/2 genes are required in the limb mesenchyme for the formation of the hindlimb. They also show that the morphological and molecular phenotypes observed in Pbx1/2 mutants are similar to those observed in Hand2 mutants. The paper presents a large amount of multi-omic analyses focusing on Pbx1 and Hand2 genome binding. The authors suggest that Pbx binds promiscuously to many genomic regions in different tissues where it acts as a pioneer factor, and that the tissue specificity is achieved by Pbx's cooperation with distinct tissue-specific transcriptional regulators. In this respect they consider the differential binding of Pbx to targets in the hindlimb, the midface and the second branchial arch, paying attention to the presence or absence in the neighboring regions of possible tissue specific collaborators like Hand2 or Hox2. The manuscript is clearly written and the experiments carefully done, however many of the conclusions are based on correlations suggested by the multi-omic data which are not independently confirmed by direct biochemical or genetic experiments. Thus, although the proposed hypothesis is credible, the authors do not present experiments directly supporting it.

➤ I find there may be a major problem with the results. The authors present careful experimental analysis with tamoxifen to find the temporal developmental window when the Pbx1/2 proteins are required for limb development. The conclusion is that hindlimbs develop normally when tamoxifen is administered at or after stage E10, suggesting Pbx function is required earlier. Despite this, all multi-omic analyses are performed using limbs at E10.5.

We thank the Reviewer for their rigor and insight. We want to clarify here that there were two main points we had to consider before embarking on all our multi-omics analyses at E10.5:

1. Conducting -omics experiments at day E9.5-E10.0 would be extremely challenging technically, if at all feasible, given that -unlike the forelimb- the hindlimb bud is barely developed by that time-point. Consequently, it would not be possible to obtain enough hindlimb tissue from E9.5-E10.0 embryos. Therefore, we chose E10.5 as the earliest time-point that allows recovery of sufficient material from murine hindlimb buds.
2. Administering tamoxifen at E9.75-10.0 most likely inactivates *Pbx1* by \geq E10.5 (as in our experience effective tamoxifen inactivation using this mouse strain takes 12-18 hrs) and subsequently the protein needs to be further cleared. Therefore, PBX is most likely required beyond E10.0, possibly even at E10.5. Accordingly, performing the multi-omics analyses at E10.5 appeared to be the most suitable time-point. In order to better clarify this point, we have now added a sentence:
"Assessment of the temporal requirements of Pbx1 during hindlimb development using the inducible Hoxb6Cre (Hoxb6^{CreERT}) deleter line³⁰, through tamoxifen injections at E8.5, E9.5, E10.0 and E10.5, respectively, showed that Pbx1/2 are required in the bud mesenchyme for hindlimb patterning prior to E10.5, considering that gene inactivation using this line takes 12-18hrs (Fig. 1I,J and Supplementary Fig. 1K)." (page 5 of the revised text).

➤ Another issue is that, even if we accept the multi-omic assays performed at those later stages are informative, the multi-omic data should have been experimentally confirmed on some of the targets.

Again, we thank the Reviewer for their rigor: this comment was fully addressed in the revised version of the manuscript. We have now conducted new experiments on wildtype and mutant hindlimb bud sections using RNAscope and choosing to assay some of the target genes within the GRN illustrated in revised Fig.9. We are excited to report that the bioinformatic predictions that led to the identification of PBX-HAND2 target genes within the GRN have now been fully validated on hindlimb bud sections of wildtype and *Pbx1/2* as well as *Hand2* mutant hindlimb buds for the target genes we selected representative of downregulated and upregulated targets, respectively. These new results are now incorporated in a new figure that we have added, Fig. 6.

➤ Thus, when in line 404 the authors claim that their "results demonstrate that promiscuous PBX transcription factors occupy a vast pool of shared genomic regions during embryonic development, but acquire restrained tissue specificity via cooperation with distinct transcriptional regulators, such as HAND2 in the

developing hindlimb.”, they are proposing a believable hypothesis that has not been demonstrated. Most of the data presented in the manuscript are sophisticated correlations that require direct molecular or biochemical confirmation by analysing any of the direct target genes detected in the study.

We thank the Reviewer for their rigor. We have now revised the manuscript by adding the necessary molecular confirmation that we obtained through validation of the perturbed target gene expression in *Pbx1/2* and *Hand2* mutant hindlimb buds by RNAscope, as explained above. These new results are now incorporated in a new figure that we have added, Fig. 6.

In addition, the new co-IP experiments illustrated in revised Fig. 3I,J provide biochemical confirmation that there is a direct interaction between PBX1 and HAND2 proteins. We conducted the co-IP experiments in two orthogonal ways: 1) using *Hand2*^{3XFlag/3XFlag} hindlimb tissue and 2) using wildtype hindlimb bud tissue at E10.5. In addition, we performed this experiment also in forelimb buds. We demonstrated that PBX1 interacts *in vivo* with HAND2 in developing limb buds. We have incorporated these new results in revised Fig. 3I,J. Uncropped immunoblots are also available to the Reviewers within the Supplementary Figures file (Uncropped blots).

- In the manuscript (most clearly in Fig. 8) there is an implicit suggestion that Hand2 autoregulates its expression in combination with Pbx. If this was true, the enhancers found in the Hand2 regulatory region could provide the perfect system to prove the hypothesis. By mutating putative Pbx-binding sites, the authors have already gone halfway into it, providing compelling evidence that one of Hand2 CRMs (mm1689) is directly regulated by Pbx1. It would have been crucial to test if this CRM is also bound by Hand2 and if the enhancer's expression requires Hand2 function in a feedback loop. If this was the case, such an element would serve to test if Hand2 increases Pbx binding to the CRM or if it collaborates as a tissue specific factor inducing localized transcription. However, in the current experiments the authors only analyse the enhancer for Pbx interactions without providing experiments to prove its possible regulation by Hand2 (neither EMSA nor directed DNA site mutagenesis has been done).

We thank again the Reviewer for their rigor and for the thoughtful suggestions that led us to further validate and strengthen our findings by conducting a large number of additional experiments. We have now re-assayed the activity of the mm1689 enhancer by mutagenizing also all HAND binding sites, similarly to the approach we adopted to mutagenize the PBX binding sites in the original manuscript. In order to be consistent, and for the assays to be truly comparable, we mutagenized the core of the *bona-fide* HAND motifs, as well as mutagenizing the same number of bases that were previously mutated within the PBX motif. Similarly, in all cases we disrupted the binding sites by substituting T/A bases into C, and C/G bases into A. We then re-assayed this mutagenized enhancer by transgenesis in the mouse and report now that mutagenesis of the HAND binding sites does not result in loss of activity of the enhancer, unlike we observed when we mutagenized the PBX binding sites. This result demonstrates that this specific enhancer (mm1689) does not require HAND2 function and is not part of a feedback loop. This finding also reinforces the concept that PBX1 might function as the upstream regulator in the cascade comprised in this GRN, even though additional layers of regulatory complexity are likely to be involved. However, this specific result does not rule out that other *Hand2* enhancers might require HAND2 function to autoregulate *Hand2* expression in combination with PBX.

- The authors genetically demonstrate that Hand2 and Pbx1 genes are collaborating during hindlimb development, however it is not clear if this is due just to Pbx1 regulating Hand2 expression, or due to direct collaborative interactions between Pbx and Hand2 proteins when activating the CRM of their direct targets.

We thank the Reviewer for the depth of this comments. A similar critique was raised by Reviewer 1. As we responded to Reviewer 1 (see above), we have addressed this important point by conducting co-immunoprecipitation experiments in embryonic limb buds *in vivo* to assess potential direct interactions between PBX and HAND2 proteins at the molecular level. We conducted the co-immunoprecipitation assays in two orthogonal ways: 1) using *Hand2*^{3XFlag/3XFlag} hindlimb bud tissue and 2) using wildtype hindlimb bud tissue at E10.5. In addition, we performed this experiment in forelimb buds. We demonstrated that PBX1 interacts *in vivo* with HAND2 in developing limb buds. We have incorporated these new results in revised Fig. 3I,J. We are now confident that there are direct collaborative interactions between PBX and HAND2 proteins when

activating the CRMs of their direct target genes. In addition, our experiments unequivocally show robust genetic interactions of *Pbx1/2* and *Hand2* during early hindlimb patterning.

Minor issues:

- Some comments could be toned down. For example, in line 489 of discussion:
“Our de novo motif analysis provides evidence that PBX1 can bind DNA without HAND2, but likely together with other TALE factors such as MEIS1/PREP1 in hindlimb buds...” It is true that this is suggested by the authors’ results, but the fact that Pbx binds DNA with TALE factors in regions that do not express Hand2 is not too surprising.

We respectfully remind the Reviewer that the regions bound by PBX1 together with other TALE factors are indeed present also in the HL, where *Hand2* is clearly expressed. Figure 3H is a *de novo* motif analysis of the HAND2 and PBX1 ChIPSeq data sets. Here, we only referred to peaks in the HL where there is a PBX1 peak without a HAND2 peak and where there is no presence of HAND2 motifs in the DNA bound by PBX1. In the revised manuscript, we have clarified the relative sentence describing these findings, which now reads:
“Our de novo motif analysis indicates that PBX1 can bind DNA without HAND2, together with other TALE factors (MEIS1/PREP1), both at promoter and intergenic/intragenic regions (Fig. 3H; *p*-value < 0.05).”
(page 8 of the revised manuscript).

- Line 512 of discussion (point 3): I find unclear what this means

In the revised manuscript, we have clarified the relative sentence as follows:
“3) PBX1 binds to largely overlapping DNA regions across multiple different embryonic tissues, including hindlimb bud, MF, and BA2.” (page 16 of the revised manuscript).

- Some references are incomplete. For example, Refs 38, 47, 59 and 69.

We thank the Reviewer for their high attention to detail. We have now manually fixed this problem, which was caused by a glitch in our bibliography system (Mendeley).

Reviewer #3 (Remarks to the Author):

This MS by Losa et al. reports on the comprehensive study of gene control by PBX1/2 and HAND2 transcription factors for early hindlimb patterning in the mouse. The question is approached through classical and conditional knock down technologies, allowing for spatial and temporal restricted loss of function, and a combination of omics approaches, including single cell and bulk RNA-seq, as well as genomic profiling for PBX1, HAND2 and histone marks, associated to ATAC-seq chromatin accessibility assays. The work is technically sound, with an elegant combination of classical and modern molecular genetics, gene expression and genomic profiling. The data is solid, clearly presented, and allows for the emergence of a detailed molecular view of the events controlling hindlimb development, which has served as an important paradigm in developmental biology. The work supports functional cooperativity of PBX1/2 and HAND2, and identifies downstream components constitutive of the GRN at work during early hindlimb patterning. The work also identifies the molecular support for a feed-forward loop whereby PBX1/2 and control the expression of HAND2.

Beyond its contribution to hindlimb development understanding, the work also impacts our understanding on how transcription factors reach specificity. This question is central to transcription factor broadly/ubiquitously expressed but that act in a spatial/temporal restricted manner. This work shows that the broadly expressed PBX1/2 proteins reach specificity in the hindlimb through functional cooperativity with HAND2, expressed in a more restricted manner within the posteriorly located hindlimb mesenchymal cells. Genetics, transcriptomics and genomics data indicate that PBX1/2 functional cooperativity relies on convergent binding to genomic regions, which resident genes are co-regulated by the PBX1/2 and HAND2.

Overall the work exploits hind limb patterning not only to clarify aspects of its morphogenesis, but also to gain insights into the mode of action of PBX proteins. The approaches taken elegantly combines modern molecular mouse genetics and omics approaches to reach solid molecular conclusions. Surprisingly, little is known on GRNs driving developmental processes and on mechanisms tuning such GRNs to different outputs. This comprehensive work in this particular context does provide significant advances that in my opinion merits publication in Nature Communications.

Below, I highlight a number of suggestions that the authors may consider to improve the manuscript.

1-The authors contextualize their work by introducing the HOX-PBX specificity paradigm. This may be rephrased as most of the data generated in this study regards PBX-HAND2 cooperativity. The end of the manuscript does compare PBX-HAND2 and PBX-HOX in the branchial arches, but this is not the core of the manuscript. I would suggest contextualizing the work as a mean to address how broadly expressed transcription factors reach specificity, which in my opinion would better suits the core of the manuscript.

We have taken the Reviewer's suggestion to heart and have refocused the paper to address how broadly expressed transcription factors reach functional specificity. For example, the first two sentences of the Abstract that previously read:

"A lingering question in developmental biology has centered on how PBX TALE transcription factors, viewed primarily as HOX co-factors, can confer functional specificity to spatially-restricted HOX proteins despite being ubiquitously distributed in vertebrate embryos. Here, using the murine hindlimb as a model, we investigate the elusive mechanisms whereby PBX homeoproteins themselves attain tissue-specific developmental functions." have now been changed as follows:

"A lingering question in developmental biology has centered on how transcription factors with widespread distribution in vertebrate embryos can perform tissue-specific functions. Here, using the murine hindlimb as a model, we investigate the elusive mechanisms whereby PBX TALE homeoproteins, viewed primarily as HOX cofactors, attain context-specific developmental roles despite ubiquitous presence in the embryo."

In addition, in the Introduction (pages 3 and 4 of the original manuscript) a few sentences have been re-written as follows:

"PBX TALE transcription factors have long been regarded as cofactors that increase the low DNA-binding specificity of HOX proteins³. However, it remains challenging to envision: 1) how PBX homeoproteins with widespread distribution in the vertebrate embryo confer functional specificity to HOX proteins that display domain-restricted localization, and 2) how they themselves attain tissue-specific developmental functions." (page 3 of the revised manuscript).

Similarly, the last sentence of the Introduction now reads:

"Broadly, this research shows how promiscuous transcription factors attain context-specific developmental functions via cooperative interactions with select cofactors that enable tissue specificity during vertebrate organogenesis." (page 4 of the revised manuscript).

Lastly, the Discussion now only tangentially touches on our previous work that reported roles of PBX transcription factors upstream of *Hox* genes and also in HOX-less domains of the embryos, demonstrating HOX-independent roles for these TALE homeoproteins, in addition to those of HOX ancillary factors. In fact, most of the revised Discussion has now been re-focused on how broadly expressed transcription factors reach functional specificity. For example, we now state:

"Our study shows that PBX TALE proteins³, characterized by a three-amino-acid loop insertion in the homeodomain, a conserved DNA-binding moiety shared by hundreds of transcription factors^{93,94}, bind indiscriminately to large numbers of CRMs shared across diverse embryonic tissues. Thus, binding of PBX alone is insufficient for context-dependent functions, whereas PBX1 binding acquires tissue-specific roles via combinatorial interactions with different cofactors, such as HAND2, which confers limb bud functionality (Fig. 9)." (pages 16-17 of the revised manuscript).

2- Functional cooperativity of PBX and HAND2 is key to the developmental system under study. When discussing the molecular bases for this cooperativity the authors consider a single scenario, that of the facilitation of HAND2 binding by PBX mediated nucleosome clearance. This would imply a temporal

sequence of PBX and HAND2 binding. Is there any indication for that? It may also be worth discussing other scenarios: do HAND2 and PBX proteins directly interact? This could be discussed based the existence/absence of relevant data concerning the role of PBX and HAND2 in other described contexts.

We thank the Reviewer for these thoughtful suggestions. To address them, we have examined whether HAND2 and PBX proteins interact directly at the molecular level by conducting co-immunoprecipitation experiments in embryonic limbs *in vivo*. We conducted the co-immunoprecipitation assays in two orthogonal ways: 1) using *Hand2*^{3XFlag/3XFlag} hindlimb bud tissue and 2) using wildtype hindlimb bud tissue at E10.5. In addition, we performed this experiment also in forelimb buds. We demonstrated that PBX1 interacts *in vivo* with HAND2 in the developing appendage. We have incorporated these new results in revised Fig. 3I,J. We are now fully confident that there are direct interactions between PBX and HAND2 proteins in the activation of their direct target genes. We have therefore added also a short paragraph in the Discussion that reads:

“Transcription factors can stabilize each other by binding to the same genomic regions in the absence of direct protein-protein interactions, thus cooperating indirectly by competing with nucleosomes^{64,79,80}. However, in this study we show that at least a fraction of the PBX1 and HAND2 transcriptional regulators are part of the same protein complexes in hindlimb buds. Such direct PBX1-HAND2 interactions were also detected in forelimb buds, where Hand2 and Pbx1 are co-expressed. However, in contrast to hindlimb buds, the functional relevance of PBX1-HAND2 protein interactions in forelimb development remains to be determined. Indeed, unlike in the hindlimb, the distal forelimb phenotype of Pbx1/2 mutants exhibits low penetrance, such that vast numbers of mutant embryos would be required in forelimb-specific genetic approaches that are beyond the scope of this study.” (page 15-16 of the revised manuscript).

Regarding the temporal sequence of *Pbx1* and *Hand2* gene activity, we assessed their expression in the lateral plate mesoderm (LPM) at E8.5 by RNAScope and demonstrated that both transcripts are already expressed at those early stages, prior budding of the hindlimb bud. Given that transcripts for both genes are co-expressed in the LPM at these very early developmental stages, and given that *Pbx1/2* expression does not precede *Hand2* expression in the LPM (see figure in 1st response to Reviewer 1), it is not possible to address the potential presence of sequential binding of PBX1 and HAND2 to chromatin *in vivo* by this indirect approach.

In light of the inconclusive results described above, we pursued a different approach to address the potential presence of a temporal sequence of PBX and HAND2 binding, as we explained in detail within our responses to Reviewer 1 (see response to Reviewer 1 above). In short, we attempted to evaluate whether PBX1 binding is affected - or not - in the absence of HAND2, in order to establish a putative sequential binding of these factors in hindlimb buds. We collected a large number of *Hand2* mutant embryos and conducted PBX1 ChIPSeq in *Hand2* mutant hindlimbs. Likely due to the low amount of mutant hindlimb tissue that could be recovered from E10.5 mutant embryos (note that hindlimb development lags behind forelimb development so that at E10.5 mouse hindlimb buds are minuscule), three different ChIPSeq experiments worked sub-optimally, despite the modifications we implemented to our established protocols to work with low amounts of starting material. Therefore, we did not feel comfortable to include these new results in the revised manuscript, as no unequivocal conclusions could be drawn from this extensive amount of additional work.

3- The reporting of a GRN mediating PBX1/2 and HAND2 is key to this work. The data reported clearly allows the identification of components of a hindlimb GRN controlled by PBX1/2 and HAND2. The authors however did not push the analysis toward the reconstruction and representation of this GRN. Data for doing so are in the MS (distinction between direct and indirect PBX and HAND2 targets, common/shared targets). These hindlimb gained knowledge could be merged with protein interaction databases to represent the backbone of a PBX/HAND2 GRN. The reason for not doing so may be cryptic, but if existing, may be discussed by the authors.

We have annotated all shared PBX1 and HAND2 target genes with respect to their spatially restricted expression in limb buds and limb phenotypes (new Suppl Table 5). This has allowed us to construct a core GRN (new Suppl Figure 8), which reveals the predicted features of the PBX1/HAND2 interactions with the target genes. One major novel finding is that in early limb buds, PBX and HAND2 co-repress genes with

roles in chondro-osteogenic differentiation and additionally positively regulate genes that inhibit this differentiation process. We have therefore added a sentence that reads:

“Notably, this analysis suggests that PBX and HAND2 co-repress genes with roles in chondro-osteogenic differentiation, while in parallel positively regulating genes that inhibit this differentiation process (Supplementary Fig. 8).” (page 11 of the revised manuscript).

4- The authors present the genetic interaction studies between PBX and HAND2 as a functional validation of the hindlimb GRN. These genetic interactions provide strong support for the functional cooperativity of PBX and HAND2, and endow functionality to many molecular observations, which are key to the present work. Yet, in my opinion, it does not per se constitute a functional validation of the newly identified GRN. Such an aim would require probing (or extracting from the existing literature) the role of some of the newly identified components, with respect to regulatory relationship and impact on hindlimb development.

We thank the Reviewer for this interesting comment. We have now addressed this point in several ways:

- 1) Regarding the point of *“newly identified components”* and whether they have a functional impact on hindlimb development, we would like to direct the Reviewer to line 316 of the original manuscript (now page 10 of the revised manuscript). Seven of the target genes identified within the hindlimb GRN are known limb regulators. We cited the specific references for each of these seven transcription factor targets, all demonstrating their critical functions in limb and/or skeletal development. The original sentence, already present when we first submitted our manuscript, reads:
“Remarkably, seven transcription factors with critical functions in limb and/or skeletal development, namely Msx148, Alx349, Lhx950, Prrx151, Zfhx452, Ets253 and Snai154,55, were found above this statistical threshold (Fig. 5A, right, red asterisks).” (page 10 of the revised manuscript).
- 2) We have further conducted new RNAscope experiments on wildtype and *Pbx1/2* as well as *Hand2* mutants hindlimb bud sections choosing select targets from those identified bioinformatically (representative of downregulated and upregulated target genes, respectively, with known roles in limb development) within the reconstructed GRN. The exciting validation results are now incorporated in a new Figure (Fig.6).
- 3) We have annotated the target genes (Suppl Table 5) and constructed a core PBX1-HAND2 target GRN (Suppl Fig.8) - for additional details please refer to our response to point 3 above.

5- The authors may explain the rationale beyond introducing conditional loss of PBX1 in a PBX2 deficient background instead of the reverse if there is one.

We would like to respectfully remind the Reviewer that *Pbx2* loss alone does not cause any detectable limb phenotypes (Selleri 2004, MCB), while loss of *Pbx1* alone results in proximal limb defects (Selleri 2001, Development). These important points were described in the Introduction (line 74 of the original manuscript; now page 3 of the revised manuscript) as follows below:

*“In contrast, compound constitutive loss-of-function of *Pbx1/2* results in multiple developmental abnormalities, including distal limb defects with loss of posterior digits⁶, and exacerbates the proximal limb phenotypes reported in *Pbx1*^{-/-} embryos⁹. As the mouse hindlimb bud lacks detectable *Pbx3*¹² and *Pbx4* -the last known *Pbx* family member- is not expressed during limb development¹³, compound loss of *Pbx1/2* achieves a PBX-null state in the developing hindlimb”.* This sentence has been left unchanged in the revised Introduction (page 3 of the revised manuscript).

In light of this knowledge, it is clear that even complete loss of *Pbx2* on a *Pbx1* heterozygous background would not yield striking limb phenotypes.

6- The authors may explain why sc-RNA seq was not employed for monitoring gene expression changes in PBX and HAND2 “mutant” conditions.

We conducted both scRNAseq and bulk RNAseq in our multi-omics study. However, we decided to use bulk RNAseq instead of scRNAseq to monitor gene expression changes in *Pbx1/2* and *Hand2* “mutant” conditions, because bulk RNAseq better captures subtle transcriptomic changes in genes that are expressed at low levels. Subtle gene expression changes would not be detected by single cell RNAseq. In addition, given that the

vast majority of cells from the hindlimb bud are mesenchymal cells (as demonstrated in our scRNAseq data sets), bulk RNAseq enabled capturing most of the gene expression changes in this subpopulation, which is the predominant subpopulation where *Hand2* and *Pbx1/2* are co-expressed.

7- To establish the PBX dependency of the newly identified HAND2 CRM, the authors probed the activity of a CRM mutated for the PBX sites. Does the CRM (WT) also lose activity in the PBX1/2 mutant context?

We thank the Reviewer for this interesting and valid suggestion. This would certainly be an interesting experiment. However, in order to answer this question we would need to make a new transgenic line with the wildtype version of mm1689 CRM on a *Pbx1/2* mutant background. This would require the generation of a quadruple mutant mouse line carrying: *HoxB6Cre*, *Pbx1* conditional allele in homozygosity, *Pbx2* null allele in homozygosity, and the transgene for the wildtype version of mm1689 CRM. We believe that the time frame for the generation of this complex mouse strain would be approximately 8 to 10 months and as such it would definitely be beyond the time frame of resubmission of this manuscript.

8- How the hindlimb GRN described here may compare to a fore limb GRN (to be described) may shortly be discussed, as it may conceptually further explain how an additional level of refinement (specificity) is achieved.

We thank the Reviewer for this thoughtful comment. The focus of the current paper is on a newly identified and newly reconstructed PBX-HAND2-dependent GRN within the hindlimb bud. Given that in the forelimb there is also expression of *Pbx3*, as detailed in the original Introduction (now page 3 of the revised manuscript), evaluating strong phenotypes in the forelimb would require the generation of quadruple mutants with loss of *Pbx1/2/3* (*Pbx1ff*; *Pbx2-/-*; *Pbx3ff*; *Hoxb6Cre*). Producing such a mutant mouse strain would require very complex genetics and complicated (as well as very rare) retrieval of quadruple mutant embryos. However, to prove that there is direct interaction of the PBX1 and HAND2 proteins at the molecular level, we performed co-IP assays not only in hindlimb buds, but also in forelimb buds. These additional experiments demonstrated that there is a direct interaction of these 2 proteins also in the forelimbs. We have incorporated these new results in revised Fig. 3I,J. In light of these additional results, the PBX-dependent GRN in the forelimb bud might be at least in part similar to that of the hindlimb bud. Accordingly, we have added a new sentence in the revised Discussion that reads as follows below:

“Transcription factors can stabilize each other by binding to the same genomic regions in the absence of direct protein-protein interactions, thus cooperating indirectly by competing with nucleosomes^{64,79,80}. However, in this study we show that at least a fraction of the PBX1 and HAND2 transcriptional regulators are part of the same protein complexes in hindlimb buds. Such direct PBX1-HAND2 interactions were also detected in forelimb buds, where Hand2 and Pbx1 are co-expressed. However, in contrast to hindlimb buds, the functional relevance of PBX1-HAND2 protein interactions in forelimb development remains to be determined. Indeed, unlike in the hindlimb, the distal forelimb phenotype of Pbx1/2 mutants exhibits low penetrance, such that vast numbers of mutant embryos would be required in forelimb-specific genetic approaches that are beyond the scope of this study.” (page 15/16 of the revised manuscript)

Reviewer #4 (Remarks to the Author):

In this work, Losa and colleagues address a conundrum in developmental genetic: how specific and precise output signals arise from complex Gene Regulatory Networks (GRN), composed of individually widespread-expressed transcription factors. In particular, the authors describe how the PBX1/2 constitutively-expressed transcription factors interplays with another regulated transcription factor, HAND2, to induce specific enhancer activities and control downstream gene transcription. The authors elegantly combine functional genetic inactivation with multi-omics approaches to precisely reconstruct the phenotypic and molecular pathways at play to control the PBX1/2-HAND2 GRN.

Overall, this study is important as it is addressing the question of epistatic relationships between factors within a model GRN. The manuscript is well written, with many cross-supportive evidence and solid techniques (e.g., the temporal requirement of *Pbx1/2* in the context of hindlimb development using *Msx2Cre* or *Hoxb6CreERT*, enhancers validation by transgenesis, mutagenization of transcription factor binding sites within the mm1689 enhancer, single-cell RNA-seq, ChIP-seq of transcription factors and

immunofluorescence). However, this reviewer has a few comments that need to be cleared by the authors and found that the intelligibility of the data presented in the figures varies enormously between the different sections of the manuscript. This let us to conclude that this manuscript needs a substantial effort to streamline and harmonize the results presented here.

Major points:

1. The authors are defining *Pbx3* as a forelimb-restricted transcription factor to justify the stronger phenotypical impact of the PBX1/PBX2 loss in hindlimbs and for using hindlimbs as a model in the present study. Yet, aside from their own study (di Giacomo 2006), *Pbx3* was never described as a forelimb specific marker gene to this reviewer knowledge. In fact, transcriptomic comparisons of fore- and hindlimbs at different developmental stages have shown low but similar expression of *Pbx3* in both tissues. Therefore, as for many other transcription factors, stronger hindlimb phenotypical outcome of the inactivation of PBX1/2 might originate from a different mechanism. Therefore, unless the authors provide strong evidence that *Pbx3* contribution in forelimbs is the cause of the weaker phenotype (i.e., triple inactivation of PBX1/2/3), this reviewer strongly encourages the authors to tone down the *Pbx3* rational in their experimental design.

We thank the Reviewer for asking further clarifications regarding this point. Bulk RNAseq assays can reveal low-level expression of any given gene. However, bulk RNAseq is most likely unable to detect expression differences that derive from the presence of specific cell populations in forelimb *versus* hindlimb (in other words, whether the same cell types in both limbs or different cells types in hindlimb *versus* forelimb are driving differences in gene expression). Therefore, we are not surprised that bulk RNAseq reveals levels of *Pbx3* gene expression in hindlimb buds purportedly similar to that of forelimb buds. It is noteworthy that, using *in situ* hybridization on sections and on whole mounts, we have shown both in DiGiacomo et al. 2006 and in Capellini et al. 2011 that the spatial expression of *Pbx3* is markedly stronger in the forelimb than in the hindlimb and is also localized to a different (and wider) anatomical domain as compared to the corresponding domain of the hindlimb. In fact, the expression of *Pbx3* in the hindlimb (HL; see figure below) is extremely weak and restricted to a modest number of proximal cells (red arrows in figure below, from Capellini et al., 2011).

[redacted]

Most notably, the patterns of *Pbx3* gene expression that we previously reported (as well as *Pbx2* and *Pbx1*, not shown here but illustrated in our publications), are fully consistent with the findings that emerged from our extensive functional studies on roles of *Pbx* family members in limb bud development. We reported a number of different experiments in which *Pbx3* was genetically deleted in the mouse model, and in all studies there were no detectable abnormalities of hindlimb development. In Rhee et al. 2004, we described that loss of *Pbx3* (*Pbx3*^{-/-}) alone does not cause abnormal hindlimb phenotypes. In Capellini et al., 2011, we reported that loss of *Pbx3* alone (*Pbx3*^{-/-}) or loss of *Pbx3* in the context of *Pbx2* loss (in *Pbx2*^{-/-}; *Pbx3*^{+/-} mutants and in *Pbx2*^{+/-}; *Pbx3*^{-/-} mutants) does not result in any detectable hindlimb phenotypes. Lastly, heterozygous loss of *Pbx3* on a *Pbx1* deficient background (*Pbx1*^{-/-}; *Pbx3*^{+/-}) does not cause hindlimb phenotypes more severe than those present in embryos with loss of *Pbx1* alone (*Pbx1*^{-/-}). In contrast, in Capellini et al. 2010 we described how loss of *Pbx3* on a *Pbx1* homozygous loss of function (i.e., *Pbx1*^{-/-}; *Pbx3*^{+/-} mutants) results in a drastic exacerbation of the forelimb defects found in single homozygous *Pbx1*^{-/-} mice. Therefore, given this strong genetic support, which derives from many years of research in our lab, we are confident that *Pbx3* is functionally a forelimb-biased gene, in the context of limb bud development as well as skeletal morphogenesis.

In order to analyze in further detail this point, we also re-analyzed bulk RNAseq datasets from a previous publication by Karen Sears’s laboratory (Transcriptomic insights into the genetic basis of mammalian limb diversity; BMC Evol Biol. 2017; 17: 86. PMC5364624). When considering the expression levels of *Pbx3*, the Reviewer is correct when stating that - at least at the mRNA level, and without considering the spatial and functional components - *Pbx3* is expressed similarly in forelimb (FL) and hindlimb (HL) buds (see figure above). However, when taking into account the spatial component, as detailed above, the interpretation becomes very different. Indeed, looking at our scRNAseq and at the newly defined mesenchymal clusters in hindlimb buds, fully consistent with all that we explained above, unlike *Pbx1* and *Pbx2*, *Pbx3* is expressed (although at low levels) only in 2 out of 8 mesenchymal clusters (1 & 4; both annotated as anterior-proximal hindlimb bud mesenchyme; see figure below). While expression data for *Pbx3* in relation to the different mesenchymal sub-clusters have not been included to the revised manuscript due to severe space constraints (only *Pbx1* and *Pbx2* data are shown in revised Suppl Fig. 2 and revised Suppl Fig. 3), we would be ready to include the figure below to the revised manuscript if the Reviewer and Editor deem it necessary.

2. The single cell analysis presented in Figure 2 is very difficult to apprehend and appear to lack important information.
 - a. The limb proximo-distal axis should be highlighted on the UMAP by using expression of known marker genes.

b. We believe that a clustering of mesenchymal cells alone could be performed so to have a more focus analysis on relevant cell types. Such a re-analysis could also be used in Figure 5.

c. The authors show 13 types of mesenchymal cluster in Figure 1D and Figure 5 without defining their identity. It is difficult to understand the rationale behind this high number and the function of each cluster. It is thus critical that the authors minimally name them. This reviewer would also propose to reduce the number of these clusters for an easier interpretation and to make it comparable with other published single cell data in the limb (see also comment “b” above). This would ease the interpretation of the large number of violin plots shown in Fig2 D/E that are presently not corresponding to the nomenclature shown in Fig2 B/C.

We would like to thank the Reviewer for these thoughtful suggestions. We have now extracted all mesenchymal cells from the hindlimb bud scRNAseq datasets and re-clustered these cells using a more robust approach, based on the evaluation of both a range of resolution values and the separation of the clusters using each value. This strategy has led us to identify a more conservative number of clusters (revised Fig. 2 and revised Suppl Fig. 2 and revised Suppl Fig. 3). Following the Reviewer’s suggestion, we also annotated these clusters reflecting the proximo-distal localization within the limb bud. We used the study by Dezanlis et al., 2020 to aid us in this new annotation. Accordingly, we have now substantially modified all figures that comprise the analysis of our scRNAseq data based on the new re-clustering (revised Fig. 2, revised Fig. 5, revised Suppl Fig. 2, revised Suppl Fig. 3, and revised Suppl Fig. 7).

Lastly, here we would like to take the opportunity to compare the old (Previous*) and new (Revised) mesenchymal clusters.

Overall, re-clustering reduced the number of mesenchymal clusters from 13 to 8. Most of the New Clusters (Revised Clusters) have a 1:1 correspondence to the original 13 clusters (Previous Clusters*), as indicated by the heat map in the upper-left corner of the figure above, which compares the number of cells in the Revised vs Previous Clusters, with only a few exceptions. New Cluster 3 is a merge of previous Clusters 6+7, while previous Cluster 12 is part of new Cluster 4. Overall, we think the present re-clustering comprising 8 clusters is overall a better classification than the previous one, with a clearer spatial distribution for each cluster. We have decided to include only the new re-clustering and to substitute all the previous panels for clarity (in revised Fig. 2, revised Fig. 5, revised Suppl Fig. 2, revised Suppl Fig. 3, and revised Suppl Fig. 7).

3. The co-expression analysis presented in Figure 5 is very interesting and clearly pinpoints the importance of cell autonomous GRNs.

a. Yet, the authors do not mention what genes are regulated in a non cell-autonomous way and impacted in other clusters than the ones co-expressing Pbx1 and Hand2. Maybe some of these genes are making sense in regard to the literature?

Respectfully, we believe that this particular point is outside the scope of the present study, which focuses on a subpopulation of posterior-proximal mesenchymal cells wherein *Pbx1/2-Hand2* are co-expressed and wherein the GRN is active. There might well be non-cell autonomous effects that impact other cell clusters. However, we believe that characterizing those would be the subject of an entire new study. That said, in the Discussion we have now added a sentence clearly stating that in this study we focused on target genes regulated in a cell-autonomous manner within the cell clusters wherein *Pbx1/2-Hand2* are co-expressed. The added sentence reads as follows:

"While the strategy employed in this study to identify a PBX-HAND2 target GRN and relevant regulatory elements points to cis-regulation, trans-regulation might also play a role within this GRN. Similarly, while shared target genes are regulated in a cell autonomous manner in the mesenchymal cell clusters that co-express Pbx1/2 and Hand2, target gene regulation in other clusters through non-cell autonomous mechanisms remains to be explored." (page 15 of the revised text).

b. Also is the Dorothea approach a confirmation of the Fig 5A analysis or does it provide new results on the co-expression of PBX/HAND2 and target genes? For now, it is presented as a new result but somehow appear redundant with the previous analysis.

We employed the Dorothea analysis as an orthogonal method to confirm the conclusions reached using Seurat, as shown in Fig. 5A. It is indeed a confirmation obtained using a published, pan-tissue, compendium of *Pbx* and *Hand2* target genes (probably less complete than our own datasets). In order to clarify this point, we have changed the specific sentence addressing this point in the Results as follows:

"An orthogonal analysis using an available computational compendium of Pbx1 and Hand2 target genes (Dorothea⁵⁶), corroborated these results (Supplementary Fig. 7D)." (page 10 of the revised manuscript).

4. The authors describe H3K27me3 as a repressive enhancer mark. This is true but, to this reviewer knowledge, only in very few specific cases. Therefore, the absence of association between PBX2/HAND2 peaks and H3K27me3 is rather biased, and it is unclear if the authors can conclude something from this analysis on enhancers. This reviewer would suggest removing this analysis from the paper or to only focus on repressed/bivalent promoter regions only.

We thank the Reviewer for this suggestion. In order to avoid confusion, we have now removed the concept of bivalent promoters that was described in the original manuscript. However, we have decided to leave the rest of the figure, as we believe that when we state that there are no significant associations with genomic regions marked by H3K27me3 repressive marks, with the exception of a fraction of promoter regions bound by PBX1 (as Suppl Fig. 5G indicates), we are providing an unbiased evaluation of our data sets. This conclusion only evaluates the presence or absence of the H3K27me3 mark segmenting the genome in promoter, intragenic and intergenic regions. Lastly, we would like to respectfully refer the Reviewer to the study from the Wysocka laboratory (Rada-Iglesias et al., 2011), which identifies potentially

bivalent enhancers (at least on a correlative basis) based on the association with the H3K27me3 mark in hESCs.

5. The transcription factor peak analysis is so far difficult to apprehend because of the following points:

a. Despite the authors presenting the binding of PBX1 genome wide as highly similar between different tissues, it appears to this reviewer that there is quite some tissue-specificity, especially looking at Fig. 7A. The authors justify here the PBX1 binding changes between cell type as being similar to the differences between CHIP replicates. Yet the cell-type specificity of PBX1 binding is then mention as relevant to HAND2/HOXA2 binding, and could therefore not be only a technical bias of CHIP, but a clear biologically relevant synergistic effect. This apparent contradiction needs to be clarified. Moreover, in the present analysis each peak is validated by replicates.

We would like to clarify this point of concern. The readout of transcription factor binding is a statistical analysis of sequence reads, where binding is a continuum from low to high binding levels (number of sequences reads relative to background). This continuum is artificially transformed into qualitative YES/NO binding by applying an arbitrary statistical cut off. In light of this process, the Venn diagram is unable to provide an absolute measure of sites where PBX binds and sites where it does not bind. Its interpretation is that relatively high-confidence peaks, i.e. in our study duplicated in each type of experiment, largely overlap across different tissues. Using lower confidence peaks (still significant) would affect the overlap across datasets, also because of technical variations between samples.

The expectation/interpretation is that the presence of the tissue-specific TF stabilizes PBX binding at selected peaks, resulting in higher PBX peaks in the corresponding tissue (rather than causing the appearance of new PBX binding events). This quantitative difference (rather than a qualitative one) is summarized in Fig. 8 J. We hope this clarifies the apparent contradiction. This point has now been further clarified in the manuscript and, to avoid controversy, we have removed the statement that PBX binding overlap across tissues is similar to a PBX duplicate within the same tissue. We have added a new sentence that reads as follows:

“Cooperative binding with HAND2 is expected to generate quantitative rather than qualitative (i.e. binding/no binding) differences in the levels of PBX occupancy at genomic sites shared across tissues. Such quantitative changes are a feature of continuous networks, in which transcription factors bind a continuum of functional and non-functional sites and regulatory specificity derives from quantitative differences in the DNA occupancy patterns⁹⁵.” (page 16-17 of the revised manuscript).

b. Despite the authors clearly showing that HAND2/HOXA2 do not bind the same regions (Fig. 7G), the overlap between their respective binding site and PBX1 cell-type specific sites is not striking (Fig. 7 D and F). This might be due to a lack of numbers in the Venn diagram or other statistical analyses, or to a lack of clarity in the author’s description of the diagrams lane 386-389.

We thank the Reviewer for raising this point. We have included the full description of the number of peaks in each comparison in Suppl Table 6. This Table also includes the percentages of each overlap calculated over the smaller dataset. We did not include the percentage of overlaps within Fig. 8 to avoid excessive crowding of the figure.

c. Finally, the example loci selected do not clearly convey the message found genome wide and developed in the text.

We thank the Reviewer for suggesting further clarification regarding the selected loci. We have now substitute the previous panels for other example loci that we believe convey the message in a more clear manner. Please see new Fig. 8, panels H and I.

6. Generally, the presentation of the conditional allele approach L130-139 is not straightforward to understand. Ideally, the authors could produce a scheme (i.e., as a Suppl figure), to illustrate, to the general audience, where and when during limb development the different Cre lines delete Pbx1.

We thank the Reviewer for suggesting the generation of an additional scheme to illustrate the activity of the different *Cre* lines used in this study, which will help those who are not familiar with limb development.

We have added a cartoon in revised Suppl Fig. 1J showing when and where the different *Cre* lines that we employed delete *Pbx1* and *Hand2* during limb bud development.

Other points

1. It is unclear to this reviewer why the authors systematically refer to GO terms in their analyses. It appears a bit as a circular argument as they work with limb tissue, limb associated gene and then, somehow obviously, find these terms associated to their analyses.

We respectfully disagree with the Reviewer on this point. We believe that referring to GO terms is critical: indeed one can see that PBX1-only bound peaks are *not* associated with limb development in the top GO categories. However, the peaks bound by PBX1-HAND2 indeed associate with limb morphogenesis and limb development in the top GO categories. We believe that the comparisons referring to the GO terms are germane to the overall message of the paper.

2. The authors need to clarify alleles nomenclature: *Hoxb6Cre/+* and *Hoxb6Cre*, *Hoxb6CreERT/+* and *Hoxb6CreERT* (see L143, L144, L149, L150, L153..).

We respectfully highlight that the nomenclature we used is correct and routinely used by the mouse genetics community. *Hoxb6^{Cre/+}* indicates a genotype where *HoxB6Cre* is present in heterozygosity, while *Hoxb6Cre* indicates the *Cre* allele that is being used. The same holds for *Hoxb6^{CreERT/+}* and *Hoxb6CreERT*.

3. L43: GRN is not defined in the Abstract section.

We have now added the full definition of “gene regulatory network” (GRN) in the revised Abstract.

4. L267: what are “complement” of limb enhancers? Are the authors talking about “repertoire” or “sets” of enhancers?

We have deleted the word "complement" to illustrate the identified *Hand2* enhancers.

5. L321: sc RNAseq and L474 scRNASeq: harmonize format

We thank the Reviewer for their rigor. We have done all edits accordingly and used consistently RNAseq.

6. L445: “Consistent with essential PBX1/2 functions in mesoderm patterning, we reported that they are required to control Polycomb and Hox gene expression in the paraxial mesoderm.” Can the authors clarify this sentence, i.e. PBX1/2 control Polycomb genes or genes that are repressed by Polycomb, such as HOX.

This sentence of the Discussion refers to findings from our laboratory reported in Capellini et al. *Developmental Biology* 2008, as referenced in the original manuscript. In that report, we established that axial skeletal patterning and hindlimb positioning are governed by *Pbx1/Pbx2* through their genetic control of both Polycomb and *Hox* expression and spatial distribution in the mesoderm. The edited sentence reads now as follows:

“Consistent with essential functions in mesoderm patterning, we reported that PBX1/2 control Polycomb and Hox gene expression in the paraxial mesoderm, which gives rise to the axial skeleton⁵.” (page 14 of the revised manuscript)

7. L460: Can the authors clarify why, with their data, they conclude that PBX/HAND2 GRN is initiating hindlimb bud patterning?

Altogether, the results obtained by the combination of tissue and temporally-controlled gene inactivation in the mouse indicates that *Pbx* is required up to E10.5 during hindlimb bud development. It is well documented in developmental biology literature that this temporal requirement coincides with very early hindlimb bud patterning.

8. L476-479: the paragraph conclusion sentence is not understandable.

We have clarified and shortened the sentence as follows:

“The spatio-temporally constrained PBX1/2-HAND2-dependent GRN consists of essential regulators of limb development that are dysregulated in Pbx1/2 and Hand2 mutant hindlimb buds.” (page 15 of the revised manuscript).

9. Fig2A: Vignettes/graphical presentations of the experiments “bulk CHIP/ATAC WT” and “bulk RNAseq” appear as dispensable to this reviewer.

We respectfully disagree with the Reviewer on this point. The Reviewer asked for the addition of a scheme illustrating the activity of the *Cre* lines that we used in our study to help those who do not work on limb development. Similarly, the schemes illustrating the *multi-omic* approaches will be useful to a general audience, especially to scientists who are devoted to classic developmental biology and do not routinely employ *-omics* approaches in their studies.

10. Fig3F-G: Not enough graphical differences between F and G panel results. A title at the top of F (biological processes) and G (mouse phenotype) would help the reader to better appreciate and discriminate those results. This comment is also valid for Fig3D-E.

We thank the Reviewer for their rigor. We have added the requested titles in the revised Fig. 3 F-G. However, we believe that Fig. 3 D-E already indicates the “significance of enrichment”. Given that the entire figure refers to ChIPseq datasets, we believe that the current annotation is self-explanatory, even without the need to read the figure legend. Also, the size of the graphs and the lettering are already very small and adding additional text would require further shrinking, thus not helping the reader.

11. Fig3I-I’-I’’: The visualization of the Both.Intergenic (I’) and Both.Promoter (I’’) subsets is not easy to comprehend (discrimination of the two boxes - solid line versus dash line borders). Maybe the authors could provide a description in the figure legend?

Trying to accommodate all the new results in this revised version of the manuscript, we have now moved the “old” panels I-I’-I’’’ from Fig. 3 to revised Suppl Fig. 5. Now the panel is much larger and, therefore, we believe that the discrimination between those two subsets of peaks is easy to identify. We have additionally enhanced the border size of the boxes to better distinguish between the solid lines and dashed lines.

12. Fig4A: H3K27ac ChIP-seq signal seems overall low in Hand2 gene desert, also very low on the putative Hand2 enhancers. Could a small fraction of Hand2-expressing cells at this developmental stage explain this result?

The Reviewer is correct. Indeed, we stated throughout the paper that *Hand2* is expressed in a small subset of posterior-proximal limb mesenchymal cells at E10.5. Therefore, the marks associated with *Hand2* gene expression, such as H3K27ac, would have an overall “diluted” signal across the entire hindlimb bud tissue that was used for the ChIP-seq experiments.

There are a few additional points to take into consideration here:

1. H2K27ac signal is mostly higher or more enriched in the promoter regions, especially if the enhancer is only active in a subset of cells used for the ChIP-seq experiments.
2. We took into account whether there was a statistically significant peak called by MACS to define those enhancers, not the “height” of the peak from the UCSC browser.

3. We only included the top panel (Fig. 4A) to help the reader visualize the location of the tested enhancers.

13. Fig4E-H: Strong differences in the X-gal staining between mm1689 HL E11.5 (Fig4E) and mm1689 WT (Fig4H). Could this be explained by difference in the transgenesis technique used (random versus enSERT transgenesis)?

We completely agree with the Reviewer. The differences in the X-gal staining are most likely explained by the difference in the transgenesis technique used (random transgenesis *versus* enSERT transgenesis for the mutagenized version of enhancer mm1689). We have added a new sentence to the figure legend to further clarify this point, which reads as follows:

“The differences in X-gal staining of the WT mm1689 enhancer are a likely consequence of the transgenesis technique used (E, random insertion; H, targeted insertion into the H11 locus by enSERT⁴⁷).”
(revised legend of Fig. 4).

14. Fig5A: Bulk RNA-seq: FC \geq 1.2, can the authors explain why they take such a low cutoff? FC \geq 1.5 is generally used in the literature.

In line with previous arguments used in this Rebuttal, the area of co-expression of *Pbx1/2* and *Hand2* (and their respective target genes) comprises only a small subset of the hindlimb bud. We have used similar approaches in previous studies looking at smaller subpopulations of cells (Gamart et al., 2021 and Laurent et al., 2017). The rationale for the use of those parameters is that some changes might be ascribed to a rather minor population of cells; therefore, we prefer to focus on the statistical significance of the differences and thus we lower the fold-change threshold not to lose many of the significant differences.

15. Fig5D: color of H3K27ac track is not matching with its name (orange versus dark green).

We thank the Reviewer for their thoroughness. We have changed this color.

16. Fig8 The message is very difficult to apprehend from the current model. Maybe having the various tissue depictions on top of one another, to compare which factors binds in each tissue could make it clearer? Moreover, the current depiction is misleading in regard to the “discordant” expressed genes (*Alx3/Msx2* and *Lhx9*). Finally, a reader could also assume from this graphical representation that all the target genes depicted here lies in cis.

This is now Fig. 9. We followed the Reviewer's suggestion and changed the representation of the genes regulated in a discordant manner (*Alx3*, *Msx2* and *Lhx9*) in the cartoon illustrating our model. While the results emerging from study aimed at identifying PBX-HAND2 target genes and regulatory elements suggest *cis*-regulation of the targets, we cannot rule out that *trans*-regulation might also play a role within the newly identified GRN. One sentence clarifying this point was added to the revised Discussion, which now reads as follows:

*“While the strategy employed in this study to identify a PBX-HAND2 target GRN and relevant regulatory elements points to cis-regulation, trans-regulation might also play a role within this GRN. Similarly, while shared target genes are regulated in a cell autonomous manner in the mesenchymal cell clusters that co-express *Pbx1/2* and *Hand2*, target gene regulation in other clusters through non-cell autonomous mechanisms remains to be explored.”*(page 15 of the revised manuscript).

REVIEWERS' COMMENTS

Reviewer #1 (Remarks to the Author):

This revised version is much improved. The authors have satisfactorily addressed my previous comments. Their conclusions are fully supported. This is an excellent, very thorough, and interesting paper that deserves to be widely read. I strongly support publication.

Reviewer #2 (Remarks to the Author):

I am satisfied with the responses to my queries and think the current manuscript can be accepted in its current form

Reviewer #3 (Remarks to the Author):

The initial review of the MS by Losa et al strongly supported publication and identified 8 points to be considered by the authors so to potentially improve the MS.

The authors have conducted an extensive revision of the MS.

This includes the introduction of substantial additional data and analysis (Fig. 3 IJ in response to point 2; Sup Table 5 and Sup Fig 8 in response to point 3; Fig 6 in response to point 4). These additions provide further insights in the PBX/Hand 2 molecular interaction, and provide a clearer representation and functional validation of the hindlimb GRN. The authors also conducted experiments to address the temporal dynamics of PBX and Hand2 genomic binding (part of point 2). Results however do not allow a clear conclusion, and were thus appropriately not included in the MS.

Changes also includes text changes so to provide a broader contextualization of the work including TF specificity (response to point 1) and hindlimb/fore limb comparison (response to point 8).

The authors also properly responded to my point 5 and 6, requesting for clarifications that could indeed be found within the initial text and literature cited.

Finally, the only point not addressed by the authors is point 7, that suggested to strengthen the conclusion on the PBX dependency of the Hand2 CRM by studying its activity in a Pbx mutant background. The authors justify this by the complexity of the genetic background to be produced. In my initial review, this was only a non-mandatory suggestion, expecting that it would be a simple experiment. I fully understand the reluctance of engaging into it especially given that it is not central to the overall solidity of the core message of the MS.

In summary, I believe the authors have fully exploited the improvement potential of my initial review. The MS has been strengthened with several respects, and in my opinion merits publication in Nature communications.

Reviewer #4 (Remarks to the Author):

The authors have satisfactorily answered the comments of this reviewer. Yet, two points still need to be finalized prior to publication.

Concerning this reviewer previous point 1 (forelimb-hindlimb Pbx3 expression differences):

>The reviewer is thankful for the very clear answer of the authors and would like to ask them to display the scRNAseq Pbx3 hindlimb expression in the final manuscript. To nail their point, it would be ideal to show it in comparison to a forelimb scRNAseq Pbx3 expression (for instance from Desanlis et al., 2020 J. Dev. Biol. or Allou et al, 2021 Nat.). Moreover, this reviewer would also then ask the authors to phrase a bit more carefully L79 and L347 that Pbx3 is weakly/differently (but not NOT) expressed in hindlimb.

Concerning this reviewer previous point 4 (H3K27me3 as a repressive enhancer mark):

>The author correctly mentioned H3K27me3 can be associated with poised enhancers as described in Rada-Iglesias original publication in 2011. Yet, they do not with repressed enhancers as they mentioned L251 of the manuscript. The authors should correct this.

Reviewers #1, 2, and 3 (Remarks to the Author):

The 3 Reviewers strongly supported publication of the manuscript in the present form.

Reviewer #4 (Remarks to the Author):

The authors have satisfactorily answered the comments of this reviewer. Yet, two points still need to be finalized prior to publication.

1) Concerning this reviewer previous point 1 (forelimb-hindlimb Pbx3 expression differences):

The reviewer is thankful for the very clear answer of the authors and would like to ask them to display the scRNAseq Pbx3 hindlimb expression in the final manuscript. To nail their point, it would be ideal to show it in comparison to a forelimb scRNAseq Pbx3 expression (for instance from Desanlis et al., 2020 J. Dev. Biol. or Allou et al., 2021 Nat.). Moreover, this reviewer would also then ask the authors to phrase a bit more carefully L79 and L347 that Pbx3 is weakly/differently (but not NOT) expressed in hindlimb.

We thank the Reviewer for their insightful suggestion and have now included, as requested, a comparison of the scRNAseq data for *Pbx3* expression in hindlimb buds from our study with the scRNAseq data for *Pbx3* expression in forelimb buds (from Desanlis et al., 2020 J. Dev. Biol.). Accordingly, we have substantially modified Supplementary Figure 3, which includes now this comparison in new Panels b and c. The additions to this figure are described in the revised manuscript as follows:

“Pbx3 was either not detected, or detected at markedly lower levels than Pbx2 in most mesenchymal clusters (Supplementary Fig. 3b,c). Notably, Pbx3 expression levels were substantially lower in hindlimb than forelimb buds (Supplementary Fig. 3b).” [page 7 of the revised manuscript]

Furthermore, as requested by the Reviewer, we have now carefully toned down all our statements regarding *Pbx3* expression throughout the manuscript. We currently describe *Pbx3* expression as follows below:

“Since mouse hindlimb buds express very low levels of Pbx3¹² and Pbx4 (the last known Pbx family member) is not expressed during limb development¹³, compound loss of Pbx1/2 likely achieves an overall PBX-null state in the developing hindlimb.” [page 3 of the revised manuscript]

“To circumvent early embryonic lethality of Pbx1/2 compound constitutive null embryos and to decipher tissue-specific Pbx functions, we conditionally inactivated Pbx1²⁶ on a Pbx2-deficient background¹⁰. This results in an overall Pbx-null state in hindlimb buds, where Pbx3 expression is extremely weak and restricted to a limited number of cells¹².” [page 5 of the revised text]

“Indeed, unlike in the hindlimb, the distal forelimb phenotype of Pbx1/2 mutants exhibits low penetrance, likely due to compensation by Pbx3 in forelimb buds.” [page 16 of the revised manuscript]

2) Concerning this reviewer previous point 4 (H3K27me3 as a repressive enhancer mark):

The author correctly mentioned H3K27me3 can be associated with poised enhancers as described in Rada-Iglesias original publication in 2011. Yet, they do not with repressed enhancers as they mentioned L251 of the manuscript. The authors should correct this.

We thank the Reviewer for their rigor and have now better described the features of the H3K27me3 mark, as requested. Accordingly, we have edited the manuscript as follows:

“Next, we integrated our sets of PBX1 and HAND2 replicated peaks with H3K27ac (associated with active enhancers) and H3K27me3 (associated with repressed promoters and poised, bivalent enhancers)^{42,43} ChIPseq profiles and ATACseq profiles (denoting open chromatin) that we generated from E10.5 hindlimb buds.” [page 8 of the revised manuscript]

We have also added the reference of Rada-Iglesias (*Nature* 2011). This is now listed as Reference #43.

We thank Reviewer 4 for their rigor and high attention to detail.